# Thermokarst lake drainage halves the temperature sensitivity of $CH_4$ release on the Qinghai-Tibet Plateau

Mei Mu[1], Cuicui Mu [1,2,3] ✉, Hebin Liu[1], Pengsi Lei[1], Yongqi Ge[1], Zhensong Zhou[1], Xiaoqing Peng[1] & Tian Ma [4]

Thermokarst lakes as hot spots of methane ($CH_4$) release are crucial for predicting permafrost carbon feedback to global warming. These lakes are suffering from serious drainage events, however, the impacts of lake drainage on $CH_4$ release remain unclear. Here, synthesizing field drilling, incubation experiments, and carbon composition and microbial communities, we reveal the temperature sensitivities ($Q_{10}$) and drivers of $CH_4$ release from drainage-affected lakes on the Qinghai-Tibet Plateau. We find that cumulative $CH_4$ release decreases with depth, where 0–30 cm-depth sediment accounts for 97% of the whole release. The $Q_{10}$ of surface sediment is 2 to 4 times higher than deep layers, but roughly 56% lower than the non-drainage lakes. The response of $CH_4$ release to warming is mainly driven by microbial communities (49.3%) and substrate availability (30.3%). Our study implies that drainage mitigates $CH_4$ release from thermokarst lakes and sheds light on crucial processes for understanding permafrost carbon projections.

Permafrost regions store ~30% of the world's soil organic carbon in approximately 15% of the Northern Hemisphere land area[1,2]. Global warming causes the thawing of permafrost and accelerates the formation, expansion, and drainage (seasonal, intermittent, or permanent) of thermokarst lakes[3–5]. These lakes and drained lake basin systems cover more than 20% of the circumpolar Northern Hemisphere permafrost regions[3,6]. Thermokarst lakes formation and expansion increase methane ($CH_4$) emission as accelerated permafrost thaw beneath and around lakes unlocks previously frozen sediments for microbial anaerobic decomposition[7,8]. Various studies have shown that thermokarst lakes serve as significant natural emission sources of $CH_4$[7,9–11], contributing 4.1–6.1 Tg $CH_4$ per year to the atmosphere[12]. It was estimated that thermokarst lakes would release 30–60 billion tonnes of carbon into the atmosphere by 2300[13], playing a crucial role in predicting permafrost carbon-climate feedback. Conversely, thermokarst lake drainage intensely alters hydrological dynamics[4],

potentially affecting $CH_4$ release to the atmosphere[3]. In the past 40 years, over 35,000 lakes have suffered from drainage events in the northern permafrost regions, half of which are thermokarst lakes (Fig. 1)[14]. However, the impact degree of thermokarst lake drainage on $CH_4$ release is poorly understood.

$CH_4$ as a powerful greenhouse gas is produced in anaerobic environments, and drainage events can significantly change $CH_4$ release from thermokarst lakes by influencing sediment moisture, carbon decomposability[15], and vegetation type markedly[16–18]. Influenced by drainage events, microbial abundance and methanogenic communities of lake sediments had great changes[19,20]. Additionally, it was shown that sediment carbon composition such as mineral-associated organic carbon (MAOC) contents increases in the drainage process due to the protection of flocculation, sorption, and co-precipitation[21]. These influencing factors are the main determinants of $CH_4$ release from thermokarst lakes[22–24]. Temperature sensitivity ($Q_{10}$)

[1]Key Laboratory of Western China's Environmental Systems (Ministry of Education), College of Earth and Environmental Sciences, Observation and research station on Eco-Environment of Frozen Ground in the Qilian Mountains, Lanzhou University, Lanzhou, China. [2]State Key Laboratory of Cryospheric Science, Northwest Institute of Eco-Environment and Resources, Chinese Academy of Sciences, Lanzhou, China. [3]Academy of Plateau Science and Sustainability, Qinghai Normal University, Xining, China. [4]State Key Laboratory of Herbage Improvement and Grassland Agroecosystems, College of Pastoral Agriculture Science and Technology, Lanzhou University, Lanzhou, China. ✉ e-mail: mucc@lzu.edu.cn

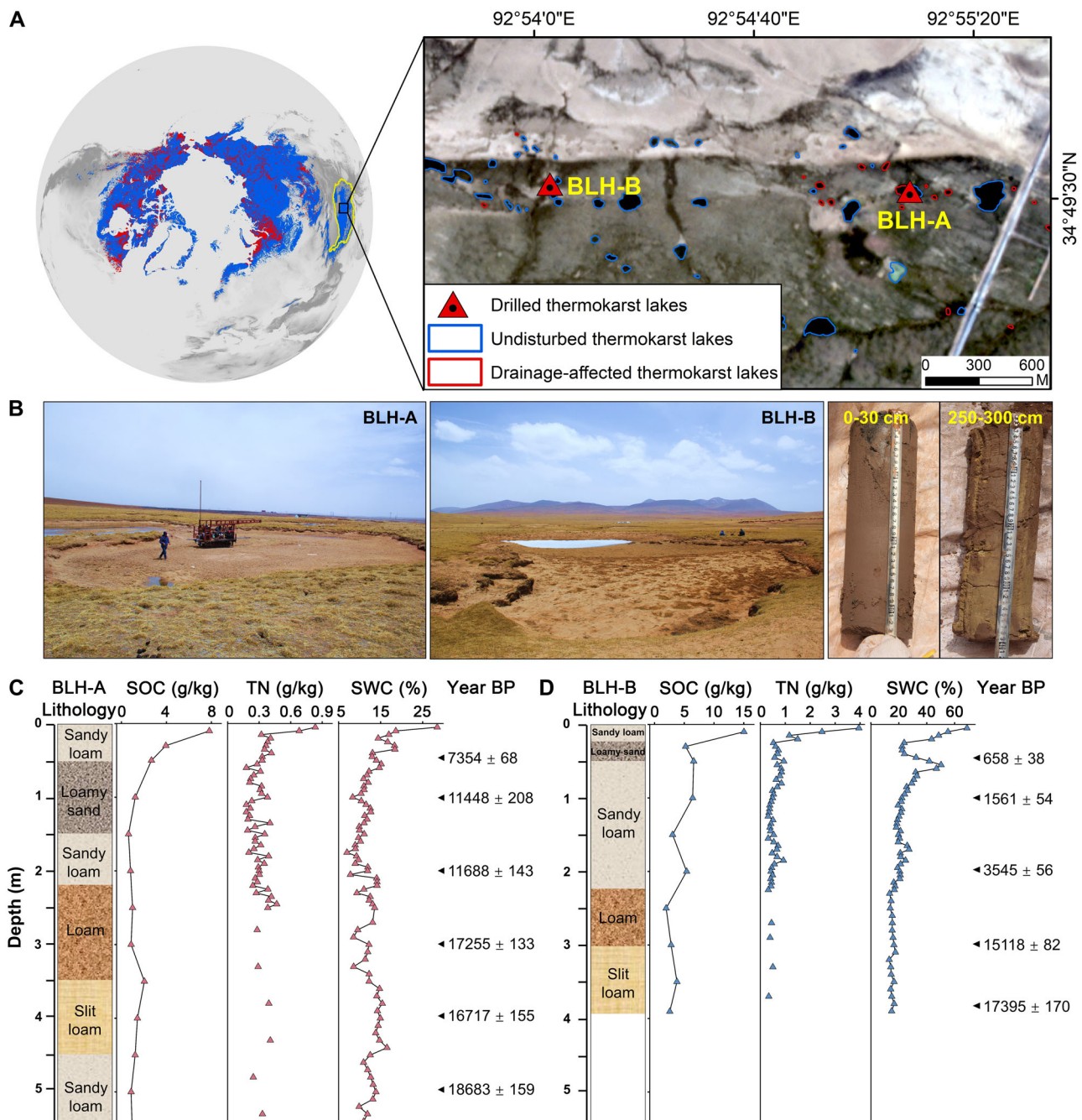

**Fig. 1 | Surveys of drainage-affected thermokarst lakes and properties of deep sediments. A** Study area with the spatial distribution of lake drainage events in the Northern Hemisphere permafrost regions[14]. The observed study area is located in the Beilu River region of QTP with the background of satellite imagery from PlanetScope 2021 (https://earth.esa.int/eogateway/missions/planetscope). The red colour represents the drainage-affected thermokarst lakes, and the blue represents the non-drainage lakes. The red triangles show the drilled thermokarst lakes at the BLH-A and BLH-B sites. **B** Field pictures of thermokarst lake BLH-A and BLH-B affected by drainage as well as the sampled deep sediment cores.
**C, D** Characteristics of two deep lake cores with the distributions of soil organic carbon (SOC) content, total nitrogen (TN) content, sediment water content (SWC), and calibrated $^{14}C$ dating and uncertainty (year BP) of SOC are shown on the right. Lithological units are identified by sediment texture. The BLH-A includes sandy loam, loamy sand, loam, and silt loam from top to deep layers. The BLH-B includes sandy loam, loamy sand, loam, and silt loam along the depth. Photos are taken by M.M. Source data are provided as a Source Data file.

represents a key parameter of biogeochemical models that reflects the response of carbon release to warming[25–27]. Quantifying the $Q_{10}$ of $CH_4$ release is thus critical to improving $CH_4$ emissions assessments of thermokarst lakes and narrowing the uncertainty of permafrost carbon-climate feedback projections[28,29]. However, the $Q_{10}$ of $CH_4$ release and its drivers in the drainage-affected thermokarst lakes remain unclear, greatly hindering the accurate assessment of permafrost carbon feedback under forthcoming climate scenarios.

The Qinghai-Tibet plateau (QTP) is the largest mountain permafrost region globally, and its warming rate is about twice that of global warming[30]. The QTP permafrost region stores a large amount of soil carbon with 12.4–25.6 billion tonnes of organic carbon in the top 2 m of soil[31]. The QTP hosts approximately 161,300 thermokarst lakes covering a total area of ~2800 km²[32]. Although rapid warming and consequent widespread permafrost thaw caused some expansion of thermokarst lakes[33], drainage events of thermokarst lakes also

emerged across the QTP over the past 40 years[34]. However, the impact of these lake drainages on $CH_4$ release is poorly understood and is also not considered in current Earth System Models, which potentially results in a rise in the uncertainty of carbon-climate feedback in the alpine permafrost regions.

To fill the knowledge gap, we drilled two 5 m-depth sediment cores in the drainage-affected thermokarst lakes on the central QTP (Fig. 1). By combining carbon composition analyses, microbial high-throughput sequencing, and radioactivity dating techniques, we conducted an anaerobic experiment over 150-day durations to present the $CH_4$ release in terms of vertical distribution, temperature sensitivity and driving mechanisms in the drainage-affected thermokarst lakes. Furthermore, by synthesizing the published $Q_{10}$ data on $CH_4$ release in the non-drainage thermokarst lakes in the same vegetation type[23], we revealed the changes of $Q_{10}$ of $CH_4$ release suffering from lake drainage. Finally, we showed the multivariate effects of influencing factors on $Q_{10}$ and relative importance using structural equation modelling (SEM) and hierarchical partitioning modelling. To our knowledge, this study first examines the dynamics of $CH_4$ release in drainage-affected thermokarst lakes in the alpine permafrost regions.

## Results and discussion
### Substrate availability and microbial communities
To reveal the vertical distribution of carbon composition and microbial communities, we conducted an analysis of multiple indices based on biochemical experiments (See methods). Sediment moisture content rapidly decreases along the depth profile and has the highest value of 14–69% in the surface 0–30 cm (Fig. 1C, D). Compared to saturated conditions in non-drainage thermokarst lakes, lake drainage significantly reduces sediment moisture contents and increases oxygen ($O_2$) availability. The calibrated ages of soil organic carbon (SOC) of deep thermokarst lake sediments are from $7,354 \pm 68$ to $18,683 \pm 159$ years BP at BLH-A site (Fig. 1C) and $658 \pm 38$ to $17,395 \pm 170$ years BP at BLH-B site (Fig. 1D), respectively. The carbon ages of sediment from the two lakes increase gradually with depth, which indicates that surface sediment has a younger carbon age and shorter time of carbon turnover. The contents of SOC and

total nitrogen (TN) decrease with depth in both drainage-affected thermokarst lakes (Fig. 1 C and D), and are significantly higher in the surface 0–30 cm layer than in the deeper sediments (>30 cm) ($p < 0.01$; Figs. 2A and S2). This phenomenon is consistent with the vertical distribution of SOC and TN in the QTP permafrost regions[35]. In contrast, the ratio of acid to aldehyde forms of vanillyl and syringyl monomers [$(Ac/Al)_V$ and $(Ac/Al)_S$] increases significantly with depth (Figure S1), and is significantly higher in deeper sediments (all $p < 0.05$; Fig. 2A). The ratio of SOC to TN (SOC/TN), particulate organic carbon to SOC (POC/SOC), and mineral-associated organic carbon to SOC (MAOC/SOC) have great variations with sediment depth (Fig. S1). Although these indices did not exhibit significant differences between surface and deeper layers ($p > 0.05$; Fig. S2), the ratios of POC/SOC and MAOC/SOC have an opposite trend. Taken together, our study demonstrates that higher substrate availability in surface sediment of thermokarst lakes is mainly dominated by POC compared to the deeper layers.

The abundances of all microbial groups, including bacterial, fungal, and actinomycetes PLFAs, significantly decrease with depth (Figs. S1 and 2B). The higher microbial abundance in surface 0–30 cm depth (all $p < 0.01$; Fig. 2B) could be attributed to higher moisture and substrate availability in surface sediment. Given the potential differences in methanogens and their metabolic pathways from different depth sediments, the functional gene abundance and community composition of methanogens were further analysed based on real-time PCR and high-throughput sequencing. The mcrA gene, which encodes a key enzyme in methanogenesis, serves as an indicator of methanogenic archaea abundance. We found that the functional gene abundance of methanogens decreases rapidly with depth (Fig. S1), with a significantly higher value in the surface than in the deeper layer ($p < 0.05$; Fig. 2B). This result indicated that there were greater potentials for $CH_4$ production in surface sediments of these two drainage-affected thermokarst lakes. The large differences in methanogens abundance depends on the corresponding disparity of substrate and sediment environmental properties between the surface and deeper layer. Regarding the methanogenic community composition of sediment from drainage-affected thermokarst lakes, there were three dominant methanogenic orders, including

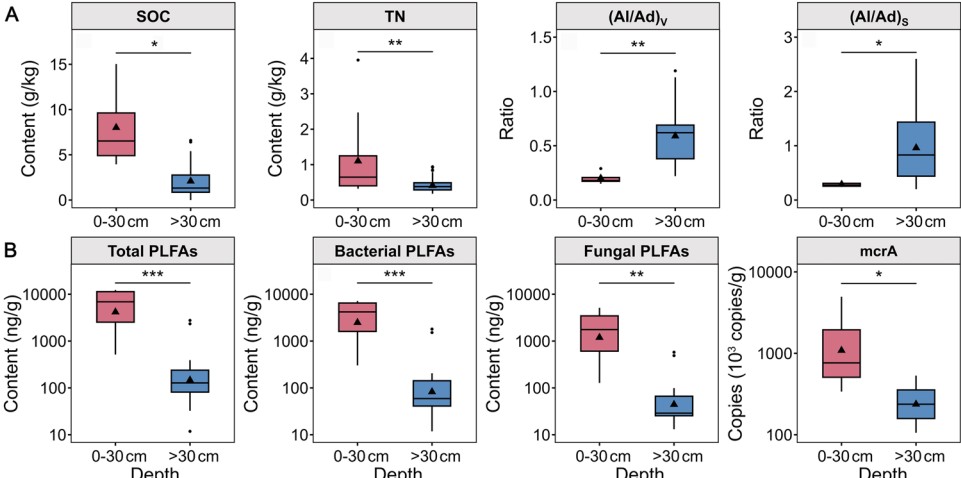

**Fig. 2 | Substrate availability and microbial abundance in surface (0–30 cm) and deep sediments (>30 cm) of the thermokarst lakes. A** The diagram shows the soil organic carbon (SOC), total nitrogen (TN), the ratio of acid to aldehyde forms of vanillyls ((Ad/Al)v), and the ratio of acid to aldehyde forms of syringyls ((Ad/Al)s). **B** depicts the microbial abundance of total, bacteria, and fungi, as well as the mcrA gene copies. Red and blue boxes indicate surface (0–30 cm) and deep sediments (>30 cm), respectively. The solid line and triangle in the box represent

the median and mean of each dataset, respectively. The upper and lower ends of boxes denote the 0.25 and 0.75 percentiles, respectively. The upper and lower whisker caps indicate the 1.5 interquartile range of upper and lower quartile, respectively. Dots outside whiskers indicate outliers. *, ** and *** indicate significant differences at $p < 0.05$, $p < 0.01$, $p < 0.001$, respectively. Source data are provided as a Source Data file.

*Methanomicrobiales*, *Methanobacteriales*, and *Methanosarcinales* (Fig. S3). This finding is consistent with results from non-drainage thermokarst lake sediments on the QTP[9] and drainage-affected permafrost tundra in the Arctic[36]. The order composition diagrams showed that *Methanomicrobiales* and *Methanobacteriales* were the most abundant methanogenic orders, representing 88-99% of all methanogens. It has been reported that the orders *Methanomicrobiales* and *Methanobacteriales* take carbon dioxide ($CO_2$) plus hydrogen ($H_2$) as substrates to produce $CH_4$[37]. The results reveal that the $CH_4$ production pathway of drainage-affected thermokarst lakes can predominantly hydrogenotrophic, which is consistent with a recent study about non-drainage thermokarst lakes on the QTP based on stable carbon isotope ($\delta^{13}C$) of $CH_4$ and $CO_2$[9]. This is possibly attributed to sediment organic matter affected by waterlogging before lake drainage is not completely degraded, providing the substrates for $CH_4$ production such as benzoate, $CO_2$, and $H_2$[38].

## Temperature sensitivity of $CH_4$ release

To estimate the potential $CH_4$ release and its $Q_{10}$ in the sediment from drainage-affected thermokarst lakes, we conducted a long-term incubation experiment (See methods). The potential $CH_4$ release rates decrease significantly with depth in thermokarst lakes of BLH-A and BLH-B, with the respective ranges of 0.002–13.50 and 0.001–65.70 μg $CH_4$ $g^{-1}$ dry sediment $d^{-1}$ (Fig. S4). Similarly, cumulative $CH_4$ release decreases significantly with depth, where 0–30 cm depth of sediment accounts for 97.2–97.7% of the whole-column $CH_4$ release (Fig. 3A, B). This result showed that $CH_4$ release of thermokarst lakes is mainly from the organic-rich surface lake sediments, which is consistent with a previous study that organic-rich mud facies accounted for 67% of whole-column $CH_4$ production in the sediment core[39]. By contrast, the cumulative $CH_4$ release was higher in BLH-B than in BLH-A, especially in the surface 0–100 cm, which may be due to large differences in moisture content, SOC and TN content, and microbial abundance in the sediment cores. Temperature increases accelerate the potential $CH_4$ release with different sediment depths, especially in the 0–30 cm layer. We find that the $Q_{10}$ of $CH_4$ release decreases with sediment depth (Fig. 3C), with the ranges of 0.41–3.58 and 0.46–3.69 in thermokarst lake BLH-A and BLH-B, respectively (Fig. S5). The highest $Q_{10}$ value occurs in the surface 0–30 cm of lake sediment (3.18 ± 0.46), which is remarkably higher than other depths (>30 cm) (all $p < 0.001$; Fig. 3D). These results corroborate the distribution of substrate availability, total microbial abundance and functional gene abundance of

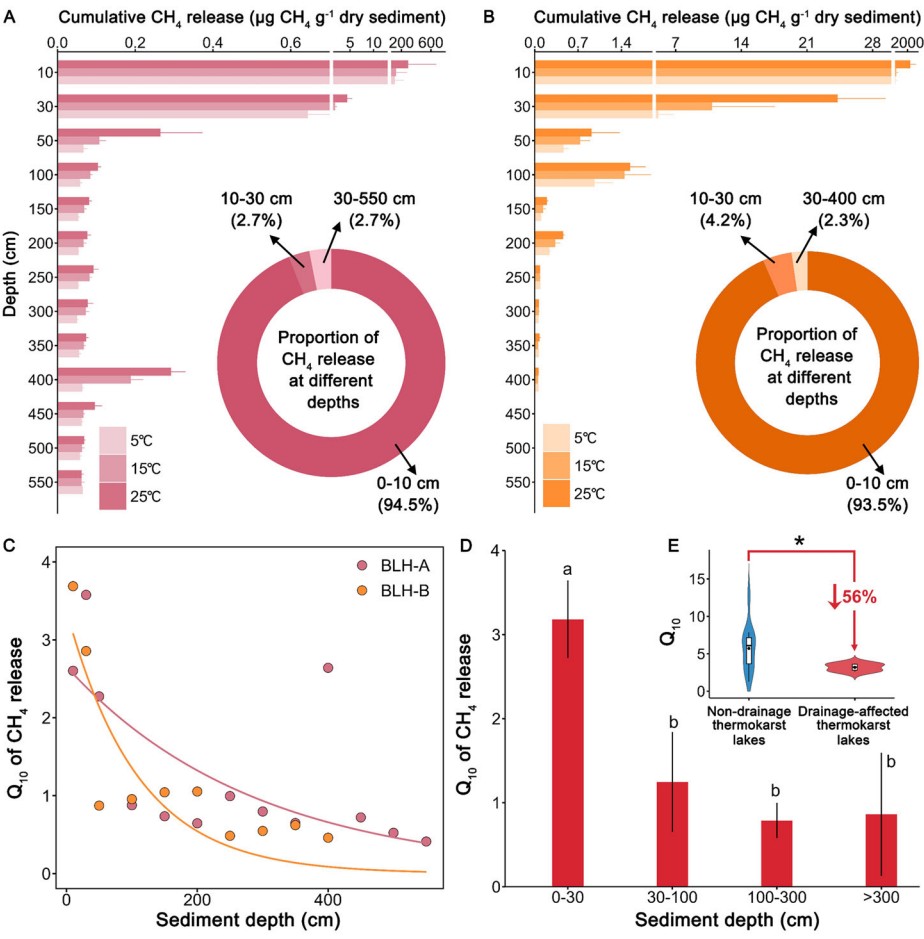

**Fig. 3 | Cumulative $CH_4$ release and its temperature sensitivity ($Q_{10}$) of thermokarst lakes. A**, **B** The bar charts show the cumulative $CH_4$ release with different sediment depths at thermokarst lakes of BLH-A and BLH-B. The columns with different colours represent the $CH_4$ release at the incubation temperatures of 5, 15, and 25 °C, respectively. Values are means ± standard errors (SE) (n = 4). The doughnut charts indicate the proportions of $CH_4$ release from sediments at depths of 0–10 cm, 10–30 cm, and >30 cm. **C** The diagram shows the changes of $Q_{10}$ with sediment depth in two thermokarst lakes affected by drainage. **D** The distribution of $Q_{10}$ values of $CH_4$ release at depths of 0–30 cm, 30–100 cm, 100–300 cm, and

>300 cm from the drainage-affected thermokarst lakes. **E** The comparison of $Q_{10}$ values of $CH_4$ release in 0–30 cm layer between the drainage-affected thermokarst lakes and non-drainage thermokarst lakes on the QTP. The data from a publication[23] and our unpublished data are shown in the supplementary materials. Values represent means ± SE. The solid line and black dots in the box represent the median and mean of each dataset, respectively. The upper and lower ends of boxes denote the 0.25 and 0.75 percentiles, respectively. The upper and lower whisker caps indicate the 1.5 interquartile range of upper and lower quartile, respectively. Source data are provided as a Source Data file.

methanogens in the vertical profiles (Fig. 2). The results suggest that the $Q_{10}$ of surface sediment in the drainage-affected thermokarst lakes is 2 to 4 times greater than deep layers (>30 cm), which provide essential parameters for permafrost carbon models.

Furthermore, to examine the effects of carbon ages on the potential $CH_4$ release and its $Q_{10}$, we analyzed the relationships of potential $CH_4$ release and $Q_{10}$ with carbon dating (See methods). The results illustrate that potential $CH_4$ release and $Q_{10}$ decreased non-linearly with carbon dating increasing, which is more pronounced in BLH-B (Fig. S6). The highest $CH_4$ release corresponds to the younger carbon in the vertical sediment profile, which is corroborated at larger study scales[40,41]. Actually, carbon dating serves as a proxy of organic matter reactivity, usually older carbon undergoes a longer period of microbial degradation, resulting in a lower reactivity than younger carbon[40]. Our finding reveals that younger carbon was a dominant source of $CH_4$ production in the drainage-affected thermokarst lakes on the QTP, which is consistent with the field observations[9].

To reveal the changes of $Q_{10}$ affected by lake drainage, we conducted a preliminary comparison between the drainage-affected and non-drainage thermokarst lakes by integrating the currently published and our unpublished $Q_{10}$ data on the QTP based on the similar incubation experiments (Table S2; See methods). Remarkably, compared with the non-drainage thermokarst lake (with an average $Q_{10}$ value of $7.20 \pm 5.76$), the $Q_{10}$ of $CH_4$ release significantly declines in the drainage-affected thermokarst lakes ($p < 0.05$; Fig. 3E). This result ties well with previous studies, showing lower water table can cause a decrease in the temperature dependence of $CH_4$ emissions in wetland ecosystems[42]. Our study quantifies the decrease of approximately 56% in the $Q_{10}$ of $CH_4$ release in the drainage-affected thermokarst lakes (Fig. 3E). This result suggests that low temperature sensitivity of drainage-affected thermokarst lakes is essential for assessing $CH_4$ emissions from thermokarst lakes. There are two possible reasons for the low temperature sensitivity. First, thermokarst lake drainage dramatically alters oxygen conditions and hydrological[3], threatening the survival of methanogens. On the one hand, thermokarst lake drainage exposes surface sediments to the atmosphere, resulting in the death of methanogens due to oxygen toxicity[43]. On the other hand, lake drainage reduces the water table depth and increases water stress on microorganisms, thereby decreasing methanogenic activity[43]. Hence, the changes in $O_2$ availability and moisture contents are important in revealing the processes of $CH_4$ production and oxidation in drainage-affected thermokarst lakes and may be key indicators for future $CH_4$ modelling. Second, lake drainage can lead to the binding of labile carbon to minerals, reducing substrate availability and thus inhibiting $CH_4$ release[44]. This explanation is consistent with that abrupt permafrost thaw resulted in significant increases in iron-bound organic carbon contents by reducing soil moisture on the QTP[45]. Abrupt permafrost thaw reduces soil moisture and improves aeration, allowing oxygen to replace Fe(III) as electron acceptors for microbial respiration and stimulating Fe(II) oxidation to Fe(III), thereby promoting the formation of iron-bound organic carbon[46,47]. Overall, our study suggests that thermokarst lake drainage halves the $Q_{10}$ of $CH_4$ release in surface sediments. However, the changes in the response of $CH_4$ release to warming during lake drainage are not considered in the simulation of permafrost carbon feedback. Therefore, our study sheds light on the crucial process for understanding the carbon-climate feedback in the changing alpine thermokarst lakes.

### Drivers of $CH_4$ release

To reveal the factors controlling the $CH_4$ release and $Q_{10}$ in drainage-affected thermokarst lakes, we established the relationships of cumulative $CH_4$ release and $Q_{10}$ with potential influencing factors, including sediment properties, substrate availability, and microbial communities (See methods) (Figs. 4, 5, and S7). The results show that the cumulative $CH_4$ release is closely related to the sediment properties and substrate availability. Specifically, cumulative $CH_4$ release positively correlates with SWC, SOC, and TN (all $p < 0.001$), and negatively correlates with clay content, (Ad/Al)v, and (Ad/Al)s (all $p < 0.05$; Fig. S7). Higher microbial abundance (all $p < 0.001$) and mcrA gene abundance ($R^2 = 0.71$, $p < 0.001$) are associated with higher $CH_4$ release (Fig. S7). Likewise, substrate availability and microbial communities are also the key predictors of $Q_{10}$ variations (Figs. 4 and 5). Specifically, the $Q_{10}$ of $CH_4$ release is positively correlated with microbial abundance, SOC, TN, and SWC (all $p < 0.01$); while is negatively correlated with (Ad/Al)v ($R^2 = 0.27$, $p < 0.05$; Fig. 4). To current knowledge, nitrogen plays an important role in $CH_4$ production because N-rich compounds are rich in proteins, which can be consumed preferentially by microbes[40]. Thus, our result suggests that TN is a crucial contributor to $CH_4$ release from thermokarst lakes. For lake drainage events, sediment moisture is severely influenced due to the changes from a waterlogged environment to terrestrial ecosystems. Low sediment moisture contents caused by lake drainage destroy anaerobic environments that hinder the $CH_4$ release[36]. Consistent with this deduction, we find faster $CH_4$ release in surface sediment with higher moisture contents (Figs. 1 and 3). Furthermore, we find that clay content is negatively correlated to the $CH_4$ release, attributing to that organic carbon is not easily accessible to microbes in clay dominant sediment. Soil organic matter can be stabilized by chemical interaction with clay minerals to form organic-mineral and also physical occlusion within microaggregates[48]. The results suggest that substrate availability, microbial communities, and sediment properties are crucial for controlling $CH_4$ release and $Q_{10}$ in drainage-affected thermokarst lakes.

To quantify the determinants and explore the mechanism influencing the $CH_4$ release and $Q_{10}$, we conducted an analysis using SEM and variation partitioning modelling. The results show that the three influencing factors together explain 93% of the variances in $CH_4$ release (Fig. 5A). Among them, substrate availability (33.4%) and microbial communities (41.1%) have significantly positive effects on $CH_4$ release based on the multiple regression models (Fig. 5B). This result is consistent with the $CH_4$ release in non-drainage thermokarst lakes on the QTP[24]. Additionally, the SEM explained 41% of the total variation in $Q_{10}$ (Fig. 5C), and substrate availability and microbial communities are responsible for 30.3% and 49.3% of the variances in $CH_4$ release, respectively (Fig. 5D). Taken together, results demonstrate that substrate availability and microbial communities are essential factors regulating the changes of $CH_4$ release and $Q_{10}$ in the drainage-affected thermokarst lakes. Our study highlights the importance of incorporating microbial communities and substrate availability into Earth System Models when predicting $CH_4$ dynamics in thermokarst lakes.

Nonetheless, our study also identifies additional knowledge gaps and perspectives. Firstly, for the drainage-affected and non-drainage thermokarst lakes, future large-scale field surveys are essential for further understanding substrate availability and microbial community composition controlling $CH_4$ production and oxidation processes. Secondly, incorporating the $CH_4$ release and $Q_{10}$ in thermokarst lakes into models is a priority for predicting future carbon-climate feedback caused by abrupt permafrost thaw. Thirdly, although our study finds that drainage events mitigate $CH_4$ release from thermokarst lakes, $CO_2$ emissions from the exposed soils could be accelerated. Therefore, comprehensive assessments of greenhouse gases in the changing thermokarst lakes are crucial for a deep understanding of the feedback of permafrost carbon to climate change.

## Methods
### Study region and field drilling
The study region was located in the Beilu River Basin of the central QTP (Fig. 1). There are two major vegetation types in this region, including alpine meadow and alpine wet meadow. The dominant plants are

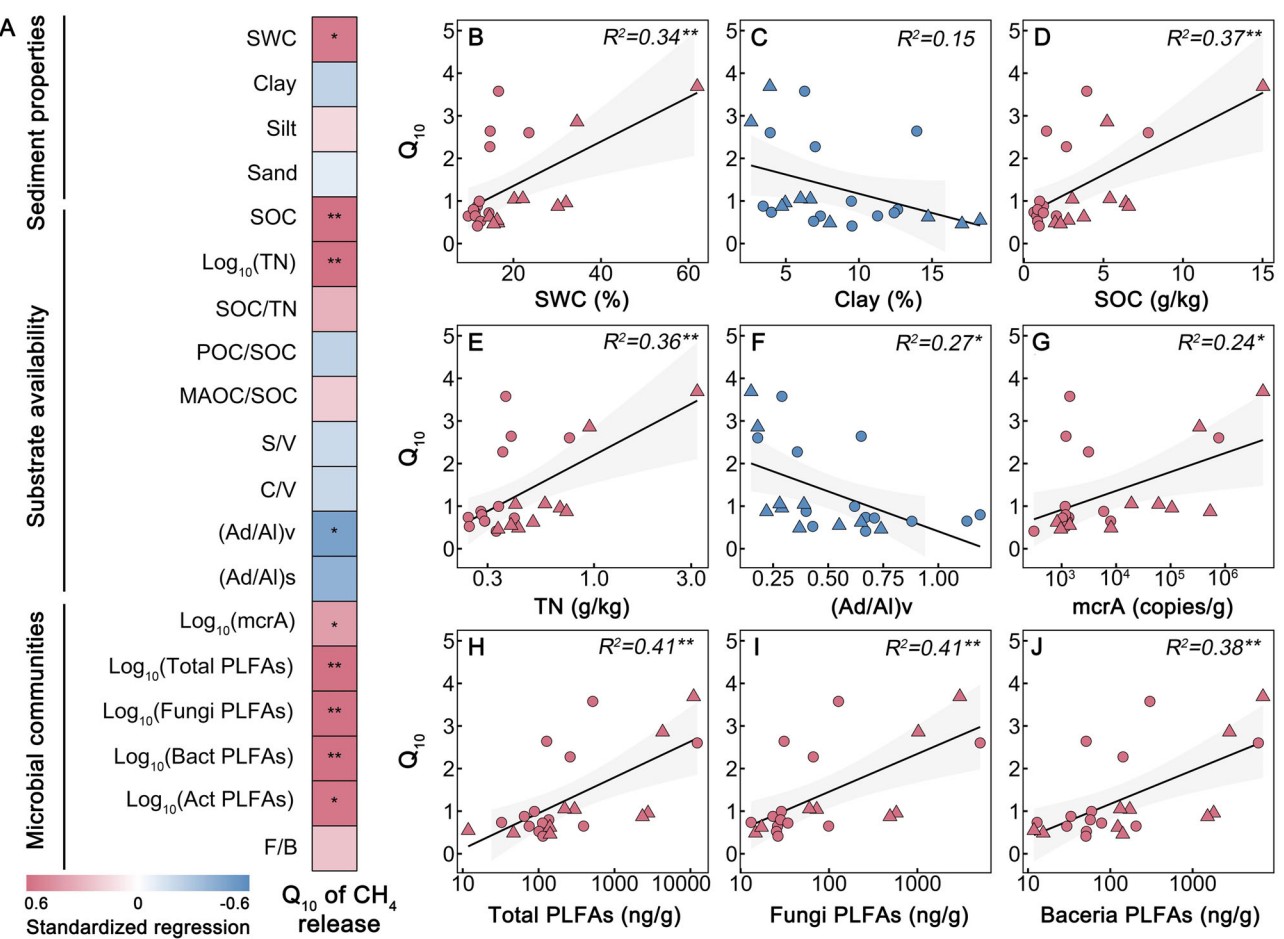

**Fig. 4 | Relationships of temperature sensitivity (Q$_{10}$) with lake sediment properties, substrate availability, and microbial communities. A** The diagram shows the standardized regression coefficient. **B–J** The correlations between Q$_{10}$ and influencing factors. SWC, sediment water content; SOC, soil organic carbon; TN, total nitrogen; POC, particulate organic carbon; MAOC, mineral-associated organic carbon; Bact, bacterial PLFAs; Act, actinomycetes PLFAs; F/B, the ratio of fungal PLFAs to bacterial PLFAs; S/V, the ratio of vanillyls to syringyls in ligninphenol; C/V, the ratio of vanillyls to cinnamyls in ligninphenol; The ratio of acid to aldehyde forms of vanillyls and syringyls [(Ad/Al)v and (Ad/Al)s]. Red and blue indicate positive and negative relationships, respectively. The solid lines and grey area represent the linear regressions and the 95% confidence interval, respectively. Circles and triangles denote BLH-A and BLH-B, respectively. R$^2$, the proportion of variance explained. *, ** and *** indicate significant correlation between Q$_{10}$ and the corresponding variable at $p < 0.05$, $p < 0.01$, $p < 0.001$, respectively. Source data are provided as a Source Data file.

*Kobresia tibetica* and *Carex spp*. This region was covered by continuous permafrost with active layer thickness ranging from 1.5 to 2.5 m and permafrost depth over 20 metres[49]. The mean annual ground temperature (at a depth of 15 m) in the study ranges from −2.0 to −0.5 °C, which is typical of a high-temperature permafrost zone[50]. Permafrost volumetric ice content is more than 30% in a large part of this region[51], leading to the widespread development of thermokarst lake. From 1969 to 2019, the number and area of thermokarst lakes increased by 159.7% and 121.5% respectively, which was driven by small lakes less than 0.5 ha in size[52]. Thermokarst lakes are characterized by small surface areas and shallow depths, which are susceptible to environmental disturbance. In this study area, the average annual temperature ranged from −6.5 to −4 °C and precipitation from 135 to 470 mm from 1955 to 2019[52]. Due to the influence of the East Asian summer monsoon, precipitation mainly occurs from May to September (Fig. S8), leading to part of thermokarst lakes experiencing large seasonal water level fluctuation annually. Field investigation and remote sensing images revealed that these lakes remain as large water bodies during the July-October, while they are drained in winter. These lakes are referred to be dry and wet cyclically as seasonal drained thermokarst lakes in this study.

Two typical seasonal drained thermokarst lakes of BLH-A and BLH-B were selected (Fig. 1A, B). The lake BLH-A (34.83°N, 92.92°E and 4648 m a.s.l) is located at alpine meadow, with lake basin area of 2530 m$^2$. The lake BLH-B (34.83°N, 92.90°E and 4,659 m a.s.l) has a basin area of 552 m$^2$ and is situated at alpine wet meadow (Table S1). The distance between the two lakes is approximately 1600 m. The vegetation within lake basins of the two seasonal drained thermokarst lakes is consistent with that around the lake. We collected two sediment cores from the bottom of two seasonal thermokarst lakes by using a drilling rig in May 2021 (Figs. 1, S9 and *Supplementary Notes*). The cores were drilled with a 10 cm diameter for the uppermost parts and an 8 cm diameter for the lower parts. The sediment core of BLH-A was 387 cm long and the sediment core of BLH-B was 555 cm long. To prevent the introduction of potential contaminants during the drilling, the outer layer of each core was scraped with autoclaved knives. The cores were labelled, packed, and stored in a −20 °C refrigerator until transported back to the laboratory. We divided the sediment cores into four equal parts: one part was subjected to air-drying for subsequent sediment physical and chemical properties measurements, another was preserved at −80 °C for subsequent DNA extraction, and the remaining two were stored at −80 °C as backup.

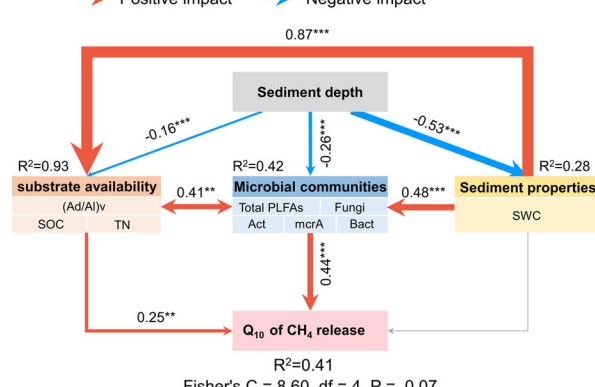

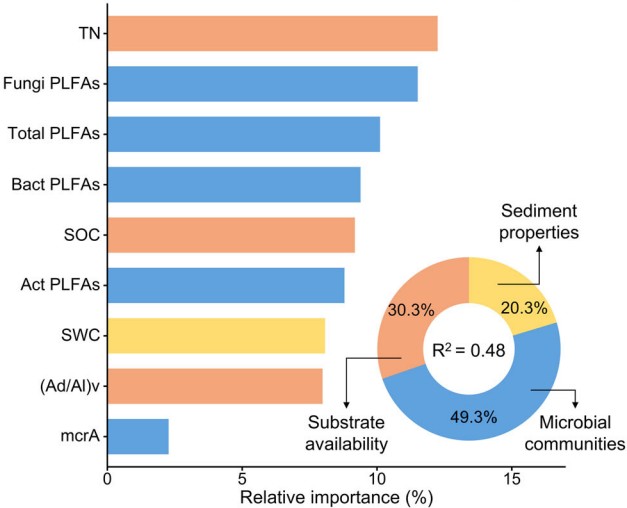

**Fig. 5 | Effects of influencing factors on CH₄ release and its temperature sensitivity (Q₁₀).** **A**, **C** Structural equation modelling (SEM) shows the multivariate effects on CH₄ release and Q₁₀. Red and blue lines indicate positive and negative relationships, respectively. Grey lines suggest insignificant paths. The width of the solid line is proportional to the correlation strength. Substrate availability includes the ratio of acid to aldehyde forms of vanillyls and syringyls [(Ad/Al)v], soil organic carbon (SOC), and total nitrogen (TN). Microbial communities are represented by total PLFAs, bacterial PLFAs (Bact), fungal PLFAs (Fungi), actinomycetes PLFAs

(Act), and the mcrA gene. Sediment properties include sediment water content (SWC) and clay content (Clay). The goodness-of-fit statistics of the model are displayed below the model. **B**, **D** Relative importance of multiple drivers in predicting CH₄ release and Q₁₀. Variation partitioning modelling evaluates the relative contributions of sediment properties, substrate availability, and microbial communities in explaining variations in CH₄ release and Q₁₀ variation. Source data are provided as a Source Data file.

## Sediment physical and chemical properties

Sediment water content was calculated by measuring the mass of the sediment before and after drying at 105 °C. Sediment texture is determined by chemically removing organic matter ($H_2O_2$ oxidation) and carbonates (HCl dissolution) using a laser particle size metre (Malvern Masterizer 2000, Malvern). Sediment total carbon (TC) and TN content were determined by high-temperature combustion using an elemental analyzer (Vario, Elementar, Hanau, Germany), and SOC content was determined by removing inorganic carbon using dilute hydrochloric acid. We calculated the weight ratio of SOC to TN, which will be referred to as the C/N ratio. The C/N indicates the degree of degradation of organic matter, with C/N decreasing with decomposition. Radiocarbon ages of sediment samples were subjected to accelerator mass spectrometry (AMS) at the Key Laboratory of Western China's Environmental Systems, Ministry of Education, Lanzhou University. The data were calibrated using the programme CALIB v 7.02 and the IntCal13 curve.

## Sediment substrate availability

We adopted SOC, TN, C/N, POC/SOC, MAOC/SOC, S/V, C/V, (Al/Ad)s, and (Al/Ad)v to characterize sediment substrate availability[53-55].

**SOC fractions.** To determine the relative contributions of MAOC and POC fraction to total SOC, we fractionated the sediments by size (53 μm) after they were fully dispersed[56]. Briefly, ~6 g air-dried sediment was shaken in 30 ml (5 g/L) sodium hexametaphosphate for 18 h to disperse the sediment completely. The dispersed sediment samples were sieved to 53 μm and rinsed with distilled water. The fraction passing through the sieve (<53 μm) was collected as MAOC, remaining on the sieve was collected as POC. After drying to constant weight in an oven at 60 °C, each fraction was analyzed for organic carbon concentration using an element analyzer (Vario, Elementar, Hanau, Germany) after acid treatment.

**Lignin phenols analysis.** The lignin phenols concentration of sediment samples was quantified by using the copper oxide (CuO) oxidation method[57]. Briefly, about 0.5–1 g freeze-dried sediment was mixed with 1 g of CuO, 0.1 g ammonium iron (II) sulfate hexahydrate, and 15 ml 2 mol/L NaOH in a tetrafluoroethylene reaction kettle. The headspace of the kettle was flushed with $N_2$ for 15 min and heated at 170 °C, for 2.5 h. The oxidation products were spiked with 400 μL ethylvanillin as recovery standard, acidified to pH<1 with 6 mol/L

hydrochloric acid, and kept in the dark for at least 1 h. After centrifugation, oxidation products were liquid-liquid extracted from the clear supernatant with ethyl acetate three times and concentrated under $N_2$ for further analysis. Lignin contents are the sum of the vanillyl (V), syringyl (S), and cinnamyl (C) monomers together[58]. Specifically, the V includes vanillin, acetovanillone, and vanillic acid. The S includes syringaldehyde, acetosyringone, and syringic acid. The C monomer was derived from the sum of p-coumaric acid and ferulic acid. The ratios of acid to aldehyde (Ad/Al) of V and S phenols were used to indicate the degree of lignin degradation and increase with lignin oxidation; The ratios of S/V and C/V were used to indicate the stability of plant substrates[59].

## Microbial abundance and community composition

We analysed the phospholipid fatty acid (PLFA) in the sediment of thermokarst lakes, which is considered a common method for assessing the microbial abundance and community composition[55,60]. The PLFAs were extracted from sediments using a chloroform-methanol-citrate buffer system following previously described procedures[61]. Before GC analysis, the samples were dissolved in hexane, and calibrated with a standard FAME solution of 19:0. A gas chromatograph and a MIDI Sherlock Microbial Identification System were used for qualitative and quantitative analysis. The PLFAs were classified as bacterial (i13:0, a13:0, i14:0, a15:0, i15:0, 15:1ω6c, i16:0, 16:1ω9c, a17:0, cy17:0, i17:0, 17:1ω6c, i18:0, 18:1ω5c, and 18:1ω7c), fungi (18:1ω9c, 18:2ω6,9c, and 18:3ω6c) and actinomycetes (16:0 Me, 17:0 Me, and 18:0 Me)[62]. Microbial community structure was assessed using the ratios of fungi to bacteria (F/B)[62].

The functional gene *mcrA* can encode methyl-coenzyme M reductase and is known as a key enzyme in methanogenesis, have been widely employed for quantification of methanogens[63,64]. In this study, to determine the abundance and community composition of methanogenic archaea, we quantified the *mcrA* gene. In detail, sediment DNA was extracted from 3 g of each freeze-dried sediment sample and then purified with a PowerMax Soil DNA Isolation Kit. DNA qualities were evaluated by using a NanoDrop ND-8000 spectrophotometer (Thermo Fisher Scientific, USA). To assess the abundance of methanogenic archaea, the quantitative real-time PCR (qPCR) was performed in a Pharmaceutical Analytics QuantStudio™ 5 Real-Time PCR System (Applied Biosystems, USA). The mcrA genes were amplified by PCR with MLf (5'- GGTGGTGTMGGATTCACACARTAYGCWACAGC -3') and MLr (5'- TTCATTGCRTAGTTWGGRTAGTT -3') as primer pairs. For each amplification, purified plasmids containing the target gene were prepared in a 10-fold dilution series for the calibration curve. Based on the calibration curve, the number of gene copies in each sample was calculated.

## Anaerobic sediment incubations

The potential $CH_4$ release rates were measured using the method described by Heslop et al[39]. The frozen samples were thawed at 4 °C overnight. From each sample, we prepared four replicates for quality control. About 10 g of fresh sediment was weighed into a 50 ml clamp-mouth anaerobic bottle and added 15 ml of ultrapure water to homogenise the sediment. The bottles were sealed with sterile butyl rubber septa and flushed with pure $N_2$ for 20 min. All bottles and butyl rubber stoppers were sterilized at 121 °C for 8 h prior to use and all manipulations were carried out in an anaerobic glove box ($N_2/H_2$, 97/3%) to exclude atmospheric $O_2$ contamination. For each sample, the replicate samples were incubated anaerobically at 5 °C, 15 °C, and 25 °C in the dark simultaneously. The incubation temperature represents the mean annual temperature, maximum summer temperature, and warming conditions at the bottom of the thermokarst lake on the QTP, respectively[65]. To measure $CH_4$ concentration, we sampled 1 mL headspace gas with a syringe and injected it into the gas chromatograph (GC, Agilent-7890A). Subsequently, 1 mL of $N_2$ was added to the bottles to

maintain air pressure equilibrium in the bottle. We measured the changes in the $CH_4$ production at nine time points during a 150-day incubation. The difference in headspace $CH_4$ concentration between two sampling time points during the incubation period was used to calculate the $CH_4$ release rate. The calculated formula is as follows[66]:

$$F = \frac{dc}{dt} \times \frac{Vh}{Ws} \times \frac{MW}{MV} \times \frac{Tst}{Tst + T} \quad (1)$$

where $F$ is the potential $CH_4$ release rate in μg $CH_4$ g⁻¹ dry sediment d⁻¹, $dc/dt$ is the change in $CH_4$ concentration in the headspace of the anaerobic bottles with incubation time. The $Vh$ is headspace volume, $Ws$ is sediment wet weight, and $MW$ and $MV$ are the molar mass and molar volume of $CH_4$ at standard conditions respectively. $Tst$ is the standard temperature (273 K), and $T$ is the incubation temperature (°C).

The $Q_{10}$ of $CH_4$ release for laboratory incubation was calculated based on the potential $CH_4$ release rate at the two incubation temperatures as follows[67]:

$$Q_{10} = \left(\frac{R_W}{R_C}\right)^{[10/T_W - T_C]} \quad (2)$$

where $R_W$ and $R_C$ are the average $CH_4$ release rate (μg $CH_4$ g⁻¹ dry sediment d⁻¹) at warmer ($T_W$) and cold ($T_C$) temperatures (°C), respectively.

## Data synthesis of $Q_{10}$ in thermokarst lakes

To show the effect of drainage on $Q_{10}$ of $CH_4$ release in thermokarst lake sediments, we synthesized the published $Q_{10}$ data of sediment $CH_4$ release in the non-drainage thermokarst lakes. Given large variations in $CH_4$ release from thermokarst lakes with different vegetation type[24], we only compiled the $Q_{10}$ data on $CH_4$ release from lake sediments located in alpine meadow and wet meadow regions on the QTP. There is limited study reporting the $Q_{10}$ of $CH_4$ release of thermokarst lake sediments[23]. We compiled our unpublish $Q_{10}$ data of non-drainage thermokarst lakes on the QTP based on similar laboratory incubation (Table S2). Considering the vertical variation of $Q_{10}$ values in this study, the surface 0-30 cm data was used for the comparison.

## Statistical analyses

All statistical analyses were performed using the software R 4.1.3[68]. Data were presented as mean ± SE and all statistical tests's significance was determined at the α = 0.05 level. Before analysis, the Quantile-Quantile Plot[69] and Levene's tests (function leveneTest)[70] were used to check the normality and homogeneity of variance for all variables, respectively. Wilcox test was conducted to examine the significance of the difference in sediment properties, substrate availability, and microbial communities with different depths using the function of *wilcox.test* in R package "stats"[68]. A paired t-test was conducted to examine the significance of the difference in $Q_{10}$ values with different depths and lake types using the function of *t.test* in the R package "rstatix"[71]. General linear models were conducted to examine the correlations of cumulative $CH_4$ release and $Q_{10}$ values with sediment properties, substrate availability, and microbial communities using the function "lm"[68]. We examined the normality and homoscedasticity of the residuals of all the linear models, and the data were logarithmically transformed when necessary. Correlations between two variables were assessed using "Pearson" correlation analysis. We also established the age-depth model for both thermokarst lake sediments using the function "lm" and "nls" in R package "stats"[68] (Fig. S10) and used the models to calculate carbon ages for other sediment depths.

To further quantify the relative contributions of sediment properties, substrate availability, and microbial communities to cumulative $CH_4$ release and $Q_{10}$, we combined a piecewise SEM, multiple linear

regression (MLR), and hierarchical partitioning to assess their relationships and relative contribution. In the initial conceptual model, we hypothesized that the microbial communities (mcrA, total PLFAs, bacterial PLFAs, fungal PLFAs, and actinomycetes PLFAs), substrate availability (SOC, TN, (Ad/Al)v and (Ad/Al)s) and sediment properties (sediment moisture and clay content) had direct effects on $CH_4$ release and $Q_{10}$. The goodness of fit of the model was evaluated using Fisher's C-statistic and the whole-model *P* value. The SEM, MLR, and hierarchical partitioning were performed using the R packages "piecewiseSEM"[72], "stats"[68], "relaimpo"[73], and "rdacca.hp"[74], respectively.

## Data availability
All data supporting the findings are available online in Supplementary information and the Figshare data repository (https://doi.org/10.6084/m9.figshare.26880526)[75]. Source data are provided in this paper.

## Code availability
Data analysis was performed using R 4.1.3. which is publicly available at https://www.r-project.org. The supporting code is provided at Figshare (https://doi.org/10.6084/m9.figshare.26892715)[76].

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

## Acknowledgements

This work was supported by the National Natural Science Foundation of China (42371132), the National Key Research and Development Programme of China (2024YFF0810900), the Gansu Science and Technology Programme (23JRRA1171, 23ZDFA017), and the Fundamental Research Funds for the Central Universities (lzujbky-2023-eyt01). We acknowledge the support of Qinghai-Beiluhe Plateau Frozen Soil Engineering Safety National Observation and Research Station.

## Author contributions

Conceptualization: C.M. and M.M. Methodology: M.M. and H.L. Investigation: M.M., H.L., P.L., Y.G., and Z.Z. Visualization: M.M. and H.L. Supervision: C.M. and X.P. Writing-original draft: M.M. and C.M. Writing-review and editing: C.M., M.M., T.M. and X.P.

## Competing interests

The authors declare no competing interests.
