## [Peer Review file · Nature Communications]

Thermokarst lake drainage halves the temperature sensitivity of CH₄ release on the Qinghai-Tibet Plateau

Corresponding Author: Professor Cuicui Mu

Version 0:

Reviewer comments:

Reviewer #1

(Remarks to the Author)

Thermokarst lake drainage halves the temperature sensitivity of CH₄ release on the Qinghai-Tibet Plateau

In this study, the authors present the results of in-depth sediment biogeochemistry analysis, potential CH₄ production, and temperature sensitivity (Q₁₀) of sediment cores from two “drainage affected” thermokarst lakes. The authors show that methane production is highest in the near surface sediments and temperature response is linked to carbon availability and *mcrA* gene copies (“microbial availability”). The authors compare their lab incubations with literature Q₁₀ data from non-drainage impacted thermokarst lakes from the same region and suggest that temperature sensitivity declines with drainage which has important implications for carbon models.

While the authors have put an incredible amount of work and data analysis into this study, I have a few major comments of concern that need to be addressed.

1. A characterization of the field sites within the main text is lacking. It is unclear if the lakes are seasonally inundated, what the vegetation species are, and how long it has been since drainage. For the study to be applicable to other permafrost regions (or models), information on the history of drainage and vegetation succession is crucial. There is also an important difference between drained lakes and seasonally drained lakes that needs to be addressed as the status of the study lakes is not clear. Especially since the results rely on only two cores, it is critical that detailed information about the drained lakes is given.

2. The authors perform anaerobic incubations to look at methane production, but it is unclear if the field conditions at the drained lakes are anaerobic in the near surface. If the near surface soils are aerobic, then the study design and application require more justification. Perhaps clarification on the “seasonal” aspects of drainage would help here. But if this is the case, then the authors should more clearly state that they are testing the potential for methane production when these drained soils are periodically inundated. The incubation results are also reported in grams of wet sediment, which makes it hard to compare across depths and sites. The authors should provide the results in grams of dry sediment.

3. The title of the manuscript and key point is that the Q₁₀ values of potential methane production from drained lakes is half of that of non-drained thermokarst lakes. Unfortunately, the manuscript lacks information about these other sites and the conditions under which Q₁₀ values were evaluated. With the current information, it is impossible for the readers to understand what these comparisons mean and it would be impossible to replicate results. Since this is the main takeaway of the paper, information on these sites and the study designs should be briefly touched on in the main text and the methods should include detailed information.

4. Throughout the manuscript, especially in the results and discussion, there is a lack of comparisons to other drained lake basin studies in permafrost regions. The manuscript could also refer to other systems that dry and wet cyclically. Some examples are below:

a. Fuchs, M., Lenz, J., Jock, S., Nitze, I., Jones, B.M., Strauss, J., Günther, F. and Grosse, G., 2019. Organic carbon and nitrogen stocks along a thermokarst lake sequence in Arctic Alaska. *Journal of Geophysical Research: Biogeosciences*, 124(5), pp.1230-1247.

b. Skeeter, J., Christen, A., Laforce, A.A., Humphreys, E. and Henry, G., 2020. Vegetation influence and environmental controls on greenhouse gas fluxes from a drained thermokarst lake in the western Canadian Arctic. *Biogeosciences*, 17(17), pp.4421-4441.

c. Loiko, S., Klimova, N., Kuzmina, D. and Pokrovsky, O., 2020. Lake drainage in permafrost regions produces variable plant communities of high biomass and productivity. *Plants*, 9(7), p.867.

d. Walter Anthony, K., Daanen, R., Anthony, P., Schneider von Deimling, T., Ping, C.L., Chanton, J.P. and Grosse, G., 2016. Methane emissions proportional to permafrost carbon thawed in Arctic lakes since the 1950s. *Nature Geoscience*, 9(9), pp.679-682.

5. The data availability statement is not enough. The original data generated by this study and also the literature review of Q10 data should be provided in a csv format and stored in an open repository. It is not enough to say the data are available in the paper and supplemental, as the data are presented in figure form with no metadata. For the study to be reproducible, it is important to make data openly available.

Minor comments:

Line 34: the term "aggravates" is not clear here as permafrost thaw can lead to thermokarst lake development and drainage.

Line 38: the phrase "current warming and dry-induced permafrost thaw" is confusing here. I am not sure what it means.

Please clarify.

Line 49: Is it not also possible that lake drainage could lead to wetland development and the presence of wetland vegetation that could lead to an increase in emissions compared to lake emissions?

Lines 51- 55: Here the authors should be clear that the relationships they are discussing were demonstrated by prior studies. As it is currently written this is not quite clear.

Lines 101-103: To my knowledge, it is quite common for sediment carbon availability to be highest at the near surface of lake sediments and wetland sediments. Some discussion of this here is warranted to provide context to the findings.

Line 149: When talking about "CH₄ release" adding the word "potential" in front is warranted for this study since the CH₄ release values are from incubations and not field studies.

Line 215: Throughout the paper "drives" should be "drivers"

Line 314: What is a seasonal thermokarst lake? The characteristics of these lakes, including age and wetting dynamics, is a crucial component of the paper and should be discussed in the main body of the manuscript.

Line 581: The data availability is not sufficient. Data files, including original study data and the literature Q10 data, need to be provided csv form in an open repository.

Reviewer #2

(Remarks to the Author)

The authors obtained two thermokarst lake sediment cores in the drainage-affected alpine permafrost area of the central Qinghai-Tibet plateau, and combined the observations with a 150-day incubation experiment to reveal the temperature sensitivity and potential drivers of CH₄ production. The mitigating effect of drainage on CH₄ release from thermokarst lakes is well known (e.g., Helsop et al. 2020), yet, the effect of drainage on thermal sensitivity (Q10 specifically) of methanogenesis and related processes has been less constrained. In this respect, the work provides interesting insights into parameterization of thermokarst processes of the central Qinghai-Tibet plateau. The consideration of microbial composition and biogeochemical processes underlying associated CH₄ release is equally interesting. The study is based on the sound methodology and in most cases provides sufficient detail for the work to be reproduced elsewhere. I believe that revision of the summarized below aspects of the manuscript would strengthen and highlight the most important conclusions.

-Introduction

31-42:

More recent and more precise references could be used;

44-45:

Please, indicate that discussed is the relevant Q10 meaning (e.g., 'here') rather than in general (in general the Q10 coefficient is a measure of the degree of temperature dependence of essentially any, e.g., chemical or biological, process).

52-55:

There are many important contributors to CH₄ release; needs to clarify that these are rather a part of the total;

55-57:

Water content/soil moisture is at least equally important a factor; especially over a much shorter temporal scale compared to that of a 10C T change potential; sediment exposure (i.e., O₂ availability) is equally important;

60:

The ref 17 seems to have a slightly different focus (also discussed in lines 61-63), while itself rather pointing to the cited ("Third Pole" (TP), air temperature increase studies by Cheng et al., 2019, Cao et al., 2018; Wu et al., 2015; Wu et al., 2010, and Kuang and Jiao, 2016; Pang et al., 2012; Yang et al., 2019); perhaps another reference would be more suitable here;

-Results

121:

Please, highlight the interpretation and significance of the observed variation.

128-129:

Please, rephrase for clarity;

130:

Also, there seems to be a very interesting set of taxonomic and biochemical data available, yet very little discussion in the text of their significance (mostly in Figures- Fig 2, 4, 5, S3, S7), nor the discussion of the associated methanogenesis pathways.

131:

Better perhaps to say - ... 'confirms' previous study;

132-134:

= different methanogenesis pathways;

-Q10

149-150:

The two cores appear to have very different soil compositions at similar depths; this may be a confounding variable in this case, perhaps worthy of some consideration and/or discussion. Also, Fig S4 - perhaps, present on a log scale? (Also Fig S6 - B);

155:

Why the 30 cm layer exactly? Do you by chance have the moisture/O₂ content profiles for the sites?

168-177:

Would be interesting to see some discussion of potentially confounding variables here (e.g., the carbon age depth profile, moisture, O₂ availability, etc.)

179-183:

check methods??

183-185:

a similar result would be expected due to lower moisture content (if this was the case in drainage affected lakes); please, clarify, and highlight the importance of your parameterization of the process, which is indeed interesting;

186-187:

There may be other (on top of the methanogens survival) factors influencing CH₄ release in these conditions, e.g., biochemical.

193-194:

Would be interesting to see some more details and clarification here.

195:

This is a very interesting conclusion (halved Q10); was it based on incubation rather than observation?; were there any in-situ measurements?

213, 432:

Please, capitalize and/or italicize SE

219-227:

Perhaps highlighting the quantitative aspects here would be more interesting than the facts themselves.

230:

TN may be either or both, contributor and/or an indicator; please clarify; please, also, rephrase 'badly' influenced;

233-234:

Please, rephrase for clarity;

235-236:

Please, clarify, whether the clay content or simply less carbon [content] was considered here;

256-260:

CH₄ release

260-264:

The presented SEM attributes 74.3% variation in Q10 to carbon availability and microbial properties; hence, what about N₂ (that seems to be included in C availability as per Fig 5), and also water and O₂ availability? Also, am I understanding correctly that these were calculated based on wet incubations in anaerobic conditions? Could you, perhaps, add some details (in the text or Supplement) to clarify your findings.

281-282:

Please, rephrase for clarity.

290-294:

A very important point indeed, especially providing many possible interlinkages between CH₄ and CO₂ emissions.

-Methods

301:

Does active layer - 1.5 to 2.5 m - affects the cores? (BLH-A - 3.87 m, BLH-B- 5.55 m; not discussed in Results)

310-312:

Please, rephrase for clarity.

315:

Please add a brief description of the drilling rig, e.g., in the Supplement.

401:

Please identify the 'previous study' here; also, did your team participate in that study?

412:

Please, specify whether CH₄ concentrations were measured directly in the headspace or sampled/extracted and then measured?

431, 433, 437, 439, 442, 446, 459, 460:

Please acknowledge/cite R and R packages (info may be found e.g., in R itself as `citation()` and `citation("package_name")` respectively)

-Supplement

Fig S2:

What is the different colours meaning?

Fig S4:

Perhaps a log scale could be considered.

Fig S6 - B:

Perhaps a log scale could be considered.

Version 1:

Reviewer comments:

Reviewer #1

(Remarks to the Author)

I appreciate the authors' thorough responses and updates to the initial reviewer suggestions. I appreciate the in-depth analysis performed by the authors, including the use of biogeochemical and microbial approaches to identify the controls on the temperature sensitivity of methane production in the sediments from these lakes. Their findings that methane production from seasonally drained thermokarst lakes has a lower temperature sensitivity than non-drainage affected thermokarst lakes is novel and would be an important contribution towards improving permafrost carbon emission models (however, see my second point below about the non-drainage affected lakes).

The authors have adequately updated their methods and now include relevant data tables and links to the data. They have also made a great effort to clarify their reasoning in many places. I have a few minor comments I hope the authors can address.

First, I still think the description of the study lakes can be improved. The thermokarst lakes are referred to as "drainage lakes", but the lakes are not permanently drained- they drain seasonally. On line 43, I suggest the authors mention that one impact of climate warming and permafrost thaw is the development of seasonally drained lakes. I also think it would be helpful to know how abundant seasonally drained lakes are compared to lakes that drain more permanently.

Second, the comparison between the q10 values from the "drainage-affected" (study lakes) and the unpublished "non-drainage thermokarst lakes" is still lacking clarity. In the methods, the authors refer to the "non-drainage lakes" as alpine meadows and wet meadows. This is confusing because this definition suggests they are permafrost wetlands and meadows, and not thermokarst lakes. If these systems are thermokarst lakes then I suggest the descriptions are updated for clarification. If these systems are not lakes, then the description and naming should be changed to highlight they are wetlands.

Finally, the shared code could be updated with more detail, including a readMe file and examples of the code with the datafiles also shared to help with reproducibility.

(Remarks on code availability)

The code does not include a readMe file. The code is for figures and the SEM, but it appears that many of the other types of analyses in the paper are left out (for example, many of the statistical analyses). The code is also not set up to run with the data files and tables provided. Instead, the code just names the data as "rawdata". This makes it difficult to reproduce the analyses. I would suggest providing more detailed code that includes the file names of the files provided so readers can more easily re-run analyses.

Reviewer #2

(Remarks to the Author)

The authors made a thorough revision of the manuscript and addressed all raised questions; thank you. This revision substantially clarifies the study design, methodology, and key findings. Few final points could, I believe, benefit from some more discussion now:

- (1) The BLH-A site appears lacking the young carbon at the surface - is there any explanation for this?
- (2) Thank you for adding more details on the methodology of incubations; perhaps adding in the supplement some data on measured CH₄ concentrations would be useful as well (i.e., undiluted, the 1 mL sample size seems quite small a volume for the reliable GC analysis); please, also note in the methods if the extracted 1 mL volume was somehow compensated in the incubation bottles (else, discuss the potentially reduced headspace volume over time);
- (3) Please, also clarify if for each sample the T was raised during the incubation, or the replicate samples were incubated at different T for 150 days simultaneously.

Minor details:

line 42-44 : please, rephrase for clarity if necessary; else, the notion seems already indicated in lines 35-36;
lines 48-52 and 52-55 : as a suggestion, I would reverse the presentation order of the two notions for segueing.

(Remarks on code availability)

I did not run the code - missing data.

REVIEWER COMMENTS

Reviewer #1 (Remarks to the Author):

Thermokarst lake drainage halves the temperature sensitivity of CH₄ release on the Qinghai-Tibet Plateau

In this study, the authors present the results of in-depth sediment biogeochemistry analysis, potential CH₄ production, and temperature sensitivity (Q₁₀) of sediment cores from two “drainage affected” thermokarst lakes. The authors show that methane production is highest in the near surface sediments and temperature response is linked to carbon availability and *mcrA* gene copies (“microbial availability”). The authors compare their lab incubations with literature Q₁₀ data from non-drainage impacted thermokarst lakes from the same region and suggest that temperature sensitivity declines with drainage which has important implications for carbon models.

While the authors have put an incredible amount of work and data analysis into this study, I have a few major comments of concern that need to be addressed.

1. A characterization of the field sites within the main text is lacking. It is unclear if the lakes are seasonally inundated, what the vegetation species are, and how long it has been since drainage. For the study to be applicable to other permafrost regions (or models), information on the history of drainage and vegetation succession is crucial. There is also an important difference between drained lakes and seasonally drained lakes that needs to be addressed as the status of the study lakes is not clear. Especially since the results rely on only two cores, it is critical that detailed information about the drained lakes is given.

2. The authors perform anaerobic incubations to look at methane production, but it is unclear if the field conditions at the drained lakes are anaerobic in the near surface. If the near surface soils are aerobic, then the study design and application require more justification. Perhaps clarification on the “seasonal” aspects of drainage would help here. But if this is the case, then the authors should more clearly state that they are testing the potential for methane production when these drained soils are periodically inundated. The incubation results are also reported in grams of wet sediment, which makes it hard to compare across depths and sites. The authors should provide the results in grams of dry sediment.

3. The title of the manuscript and key point is that the Q₁₀ values of potential methane production from drained lakes is half of that of non-drained thermokarst lakes. Unfortunately, the manuscript lacks information about these other sites and the conditions under which Q₁₀ values were evaluated. With the current information, it is impossible for the readers to understand what these comparisons mean and it would be impossible to replicate results. Since this is the main takeaway of the paper,

information on these sites and the study designs should be briefly touched on in the main text and the methods should include detailed information.

4. Throughout the manuscript, especially in the results and discussion, there is a lack of comparisons to other drained lake basin studies in permafrost regions. The manuscript could also refer to other systems that dry and wet cyclically. Some examples are below:

a. Fuchs, M., Lenz, J., Jock, S., Nitze, I., Jones, B.M., Strauss, J., Günther, F. and Grosse, G., 2019. Organic carbon and nitrogen stocks along a thermokarst lake sequence in Arctic Alaska. *Journal of Geophysical Research: Biogeosciences*, 124(5), pp.1230-1247.

b. Skeeter, J., Christen, A., Laforce, A.A., Humphreys, E. and Henry, G., 2020. Vegetation influence and environmental controls on greenhouse gas fluxes from a drained thermokarst lake in the western Canadian Arctic. *Biogeosciences*, 17(17), pp.4421-4441.

c. Loiko, S., Klimova, N., Kuzmina, D. and Pokrovsky, O., 2020. Lake drainage in permafrost regions produces variable plant communities of high biomass and productivity. *Plants*, 9(7), p.867.

d. Walter Anthony, K., Daanen, R., Anthony, P., Schneider von Deimling, T., Ping, C.L., Chanton, J.P. and Grosse, G., 2016. Methane emissions proportional to permafrost carbon thawed in Arctic lakes since the 1950s. *Nature Geoscience*, 9(9), pp.679-682.

5. The data availability statement is not enough. The original data generated by this study and also the literature review of Q10 data should be provided in a csv format and stored in an open repository. It is not enough to say the data are available in the paper and supplemental, as the data are presented in figure form with no metadata. For the study to be reproducible, it is important to make data openly available.

Minor comments:

Line 34: the term “aggravates” is not clear here as permafrost thaw can lead to thermokarst lake development and drainage.

Line 38: the phrase “current warming and dry-induced permafrost thaw” is confusing here. I am not sure what it means. Please clarify.

Line 49: Is it not also possible that lake drainage could lead to wetland development and the presence of wetland vegetation that could lead to an increase in emissions compared to lake emissions?

Lines 51- 55: Here the authors should be clear that the relationships they are discussing were demonstrated by prior studies. As it is currently written this is not quite clear.

Lines 101-103: To my knowledge, it is quite common for sediment carbon availability to be highest at the near surface of lake sediments and wetland sediments. Some

discussion of this here is warranted to provide context to the findings.

Line 149: When talking about “CH₄ release” adding the word “potential” in front is warranted for this study since the CH₄ release values are from incubations and not field studies.

Line 215: Throughout the paper “drives” should be “drivers”

Line 314: What is a seasonal thermokarst lake? The characteristics of these lakes, including age and wetting dynamics, is a crucial component of the paper and should be discussed in the main body of the manuscript.

Line 581: The data availability is not sufficient. Data files, including original study data and the literature Q10 data, need to be provided csv form in an open repository.

Reviewer #2 (Remarks to the Author):

The authors obtained two thermokarst lake sediment cores in the drainage-affected alpine permafrost area of the central Qinghai-Tibet plateau, and combined the observations with a 150-day incubation experiment to reveal the temperature sensitivity and potential drivers of CH₄ production. The mitigating effect of drainage on CH₄ release from thermokarst lakes is well known (e.g., Helsop et al. 2020), yet, the effect of drainage on thermal sensitivity (Q10 specifically) of methanogenesis and related processes has been less constrained. In this respect, the work provides interesting insights into parameterization of thermokarst processes of the central Qinghai-Tibet plateau. The consideration of microbial composition and biogeochemical processes underlying associated CH₄ release is equally interesting. The study is based on the sound methodology and in most cases provides sufficient detail for the work to be reproduced elsewhere. I believe that revision of the summarized below aspects of the manuscript would strengthen and highlight the most important conclusions.

-Introduction

31-42: More recent and more precise references could be used;

44-45: Please, indicate that discussed is the relevant Q10 meaning (e.g., 'here') rather than in general (in general the Q10 coefficient is a measure of the degree of temperature dependence of essentially any, e.g., chemical or biological, process).

52-55: There are many important contributors to CH₄ release; needs to clarify that these are rather a part of the total;

55-57: Water content/soil moisture is at least equally important a factor; especially over a much shorter temporal scale compared to that of a 10C T change potential; sediment exposure (i.e., O₂ availability) is equally important;

60: The ref 17 seems to have a slightly different focus (also discussed in lines 61-63), while itself rather pointing to the cited ("Third Pole" (TP), air temperature increase studies by Cheng et al., 2019, Cao et al., 2018; Wu et al., 2015; Wu et al., 2010, and Kuang and Jiao, 2016; Pang et al., 2012; Yang et al., 2019); perhaps another reference would be more suitable here;

-Results

121: Please, highlight the interpretation and significance of the observed variation.

128-129: Please, rephrase for clarity;

130: Also, there seems to be a very interesting set of taxonomic and biochemical data available, yet very little discussion in the text of their significance (mostly in Figures- Fig 2, 4, 5, S3, S7), nor the discussion of the associated methanogenesis pathways.

131: Better perhaps to say - ... 'confirms' previous study;

132-134: different methanogenesis pathways;

-Q10

149-150: The two cores appear to have very different soil compositions at similar depths; this may be a confounding variable in this case, perhaps worthy of some consideration and/or discussion. Also, Fig S4 - perhaps, present on a log scale? (Also Fig S6 - B);

155: Why the 30 cm layer exactly? Do you by chance have the moisture/O₂ content profiles for the sites?

168-177: Would be interesting to see some discussion of potentially confounding variables here (e.g., the carbon age depth profile, moisture, O₂ availability, etc.)

179-183: check methods??

183-185: a similar result would be expected due to lower moisture content (if this was the case in drainage affected lakes); please, clarify, and highlight the importance of your parameterization of the process, which is indeed interesting;

186-187: There may be other (on top of the methanogens survival) factors influencing CH₄ release in these conditions, e.g., biochemical.

193-194: Would be interesting to see some more details and clarification here.

195: This is a very interesting conclusion (halved Q10); was it based on incubation rather than observation?; were there any in-situ measurements?

213, 432: Please, capitalize and/or italicize SE

219-227: Perhaps highlighting the quantitative aspects here would be more interesting than the facts themselves.

230: TN may be either or both, contributor and/or an indicator; please clarify; please, also, rephrase 'badly' influenced;

233-234: Please, rephrase for clarity;

235-236: Please, clarify, whether the clay content or simply less carbon [content] was considered here;

256-260: CH₄ release

260-264: The presented SEM attributes 74.3% variation in Q10 to carbon availability and microbial properties; hence, what about N₂ (that seems to be included in C availability as per Fig 5), and also water and O₂ availability? Also, am I understanding correctly that these were calculated based on wet incubations in anaerobic conditions? Could you, perhaps, add some details (in the text or Supplement) to clarify your findings.

281-282: Please, rephrase for clarity.

290-294: A very important point indeed, especially providing many possible interlinkages between CH₄ and CO₂ emissions.

-Methods

301: Does active layer - 1.5 to 2.5 m - affects the cores? (BLH-A - 3.87 m, BLH-B- 5.55 m; not discussed in Results)

310-312: Please, rephrase for clarity.

315: Please add a brief description of the drilling rig, e.g., in the Supplement.

401: Please identify the 'previous study' here; also, did your team participate in that study?

412: Please, specify whether CH₄ concentrations were measured directly in the headspace or sampled/extracted and then measured?

431, 433, 437, 439, 442, 446, 459, 460: Please acknowledge/cite R and R packages (info may be found e.g., in R itself as citation() and citation("package_name") respectively)

-Supplement

Fig S2: What is the different colours meaning?

Fig S4: Perhaps a log scale could be considered.

Fig S6 - B: Perhaps a log scale could be considered.

Response to Reviewers

Reviewer #1 (Remarks to the Author):

Thermokarst lake drainage halves the temperature sensitivity of CH₄ release on the Qinghai-Tibet Plateau

In this study, the authors present the results of in-depth sediment biogeochemistry analysis, potential CH₄ production, and temperature sensitivity (Q₁₀) of sediment cores from two “drainage affected” thermokarst lakes. The authors show that methane production is highest in the near surface sediments and temperature response is linked to carbon availability and mcrA gene copies (“microbial availability”). The authors compare their lab incubations with literature Q₁₀ data from non-drainage impacted thermokarst lakes from the same region and suggest that temperature sensitivity declines with drainage which has important implications for carbon models.

While the authors have put an incredible amount of work and data analysis into this study, I have a few major comments of concern that need to be addressed.

Response: Thank you for your recognition and suggestions on our study. These comments are all valuable and helpful for revising and improving our paper. The detailed revisions in the paper are as follows:

1. A characterization of the field sites within the main text is lacking. It is unclear if the lakes are seasonally inundated, what the vegetation species are, and how long it has been since drainage. For the study to be applicable to other permafrost regions (or models), information on the history of drainage and vegetation succession is crucial. There is also an important difference between drained lakes and seasonally drained lakes that needs to be addressed as the status of the study lakes is not clear. Especially since the results rely on only two cores, it is critical that detailed information about the drained lakes is given.

Response: It is really true as you suggest that the characterization of field sites is important for the study. According to your suggestion, we have provided information about the characterization of two thermokarst lakes in the Methods.

Line 355: *The dominant plants are Kobresia tibetica and Carex spp.*

Lines 364-379: *In this study area, the average annual temperature ranged from -6.5 to -4°C and precipitation from 135 to 470 mm from 1955 to 2019⁵⁰. Due to the influence of the East Asian summer monsoon, precipitation mainly occurs from May to September (Fig. S8), leading to part of thermokarst lakes experiencing large seasonal water level fluctuation annually. Field investigation and remote sensing images revealed that these lakes remain as large water bodies during the July-October, while they are drained in winter. These lakes are referred to be dry and wet cyclically as seasonal drained thermokarst lakes in this study.*

Two typical seasonal drained thermokarst lakes of BLH-A and BLH-B were selected (Fig. 1A and B). The lake BLH-A (34.83°N, 92.92°E and 4,648 m a.s.l) is located at alpine meadow, with lake basin area of 2,530 m². The lake BLH-B (34.83°N,

92.90°E and 4,659 m a.s.l) has a basin area of 552 m² and is situated at alpine wet meadow (Table S1). The distance between the two lakes is approximately 1,600 m. The vegetation within lake basins of the two seasonal drained thermokarst lakes is consistent with that around the lake.

Fig. S8 Monthly average temperature and precipitation in the study area from the Wudaoliang meteorological station during 1990-2020. The solid line and triangle in the box represent the median and mean of each dataset, respectively. The upper and lower ends of boxes denote the 0.25 and 0.75 percentiles, respectively. The upper and lower whisker caps indicate the 1.5 interquartile range of upper and lower quartile, respectively. Dots outside whiskers indicate outliers.

2. The authors perform anaerobic incubations to look at methane production, but it is unclear if the field conditions at the drained lakes are anaerobic in the near surface. If the near surface soils are aerobic, then the study design and application require more justification. Perhaps clarification on the “seasonal” aspects of drainage would help here. But if this is the case, then the authors should more clearly state that they are testing the potential for methane production when these drained soils are periodically inundated. The incubation results are also reported in grams of wet sediment, which

makes it hard to compare across depths and sites. The authors should provide the results in grams of dry sediment.

Response: We appreciate your great comments. The thermokarst lakes in our study are seasonal drained thermokarst lakes. According to your above comment, we have added the detailed information about the study region and two thermokarst lakes **[lines 355, 364-379]**. Thus, our study design and application are appropriate to answer our concerned scientific question more clearly.

Additionally, all figures are redrawn and presented as the grams of dry sediment in our revised manuscript. Meanwhile, we have modified the relevant description. For example:

Lines 26-27: *We find that cumulative CH₄ release decreases with depth, where 0-30 cm-depth sediment accounts for 97% of the whole release.*

Lines 28-30: *....., but roughly 56% lower than the non-drainage lakes. The response of CH₄ release to warming is mainly driven by microbial communities (49.3%) and substrate availability (30.3%).*

Lines 179-183: *The potential CH₄ release rates decrease significantly with depth in thermokarst lakes of BLH-A and BLH-B, with the respective ranges of 0.002-13.50 and 0.001-65.70 μg CH₄ g⁻¹ dry sediment d⁻¹ (Fig. S4). Similarly, cumulative CH₄ release decrease significantly with depth, where 0-30 cm-depth of sediment accounts for 97.2-97.7% of the whole-column CH₄ release (Figs. 3A, B).*

Lines 192-193: *....., with the ranges of 0.41-3.58 and 0.46-3.69 in thermokarst lake BLH-A and BLH-B, respectively (Fig. S5).*

Lines 311-318: *The results show that the three influencing factors together explain 93% of the variances in CH₄ release (Fig. 5A). Among them, substrate availability (33.4%) and microbial communities (41.1%) have significantly positive effects on CH₄ release based on the multiple regression models (Fig. 5B). This result is consistent with the CH₄ release in non-drainage thermokarst lakes on the QTP²⁷. What's more, the SEM explained 41% of the total variation in Q₁₀ (Fig. 5C), and substrate availability and microbial communities are responsible for 30.3% and 49.3% of the variances in CH₄ release, respectively (Fig. 5D).*

Fig. 3. Cumulative CH_4 release and its temperature sensitivity (Q_{10}) of thermokarst lakes. (A-B) The bar charts show the cumulative CH_4 release with different sediment depths at thermokarst lakes of BLH-A and BLH-B. The columns with different colours represent the CH_4 release at the incubation temperatures of 5, 15, and 25°C, respectively. Values are means \pm standard errors (SE) ($n=4$). The doughnut charts indicate the proportions of CH_4 release from sediments at depths of 0-10 cm, 10-30 cm, and >30 cm. **(C)** The diagram shows the changes of Q_{10} with sediment depth in two thermokarst lakes affected by drainage. **(D)** The distribution of Q_{10} values of CH_4 release at depths of 0-30 cm, 30-100 cm, 100-300 cm, and >300 cm from the drainage-affected thermokarst lakes. **(E)** The comparison of Q_{10} values of CH_4 release in 0-30 cm layer between the drainage-affected thermokarst lakes and non-drainage thermokarst lakes on the QTP. The data from a publication²⁶ and our unpublished data are shown in the supplementary materials. Values represent means \pm SE. The solid line and black dots in the box represent the median and mean of each dataset, respectively. The upper and lower ends of boxes denote the 0.25 and 0.75 percentiles, respectively. The upper and lower whisker caps indicate the 1.5 interquartile range of upper and lower quartile, respectively.

Fig. 4. Relationships of temperature sensitivity (Q_{10}) with lake sediment properties, substrate availability, and microbial communities. (A) The diagram shows the standardized regression coefficient. **(B-J)** The correlations between Q_{10} and influencing factors. SWC, sediment water content; SOC, soil organic carbon; TN, total nitrogen; POC, particulate organic carbon; MAOC, mineral-associated organic carbon; Bact, bacterial PLFAs; Act, actinomycetes PLFAs; F/B, the ratio of fungal PLFAs to bacterial PLFAs; S/V, the ratio of vanillyls to syringyls in ligninphenol; C/V, the ratio of vanillyls to cinnamyls in ligninphenol; The ratio of acid to aldehyde forms of vanillyls and syringyls [(Ad/Al)v and (Ad/Al)s]. The solid lines and grey area represent the linear regressions and the 95% confidence interval, respectively. Circles and triangles denote BLH-A and BLH-B, respectively. R^2 , the proportion of variance explained. *, ** and *** indicate significant correlation between Q_{10} and the corresponding variable at $p < 0.05$, $p < 0.01$, $p < 0.001$, respectively.

Fig. 5. Effects of influencing factors on CH₄ release and its temperature sensitivity (Q₁₀). (A, C) Structural equation modelling (SEM) shows the multivariate effects on CH₄ release and Q₁₀. Red and blue lines indicate positive and negative relationships, respectively. Grey lines suggest insignificant paths. The width of the solid line is proportional to the correlation strength. Substrate availability includes the ratio of acid to aldehyde forms of vanillyls and syringyls [(Ad/Al)v], soil organic carbon (SOC), and total nitrogen (TN). Microbial communities are represented by total PLFAs, bacterial PLFAs (Bact), fungal PLFAs (Fungi), actinomycetes PLFAs (Act), and the *mcrA* gene. Sediment properties include sediment water content (SWC) and clay content (Clay). The goodness-of-fit statistics of the model are displayed below the model. (B, D) Relative importance of multiple drivers in predicting CH₄ release and Q₁₀. Variation partitioning modelling evaluates the relative contributions of sediment properties, substrate availability and microbial communities in explaining variations in CH₄ release and Q₁₀ variation.

Fig. S4. CH₄ release rates obtained from anaerobic incubations. (A-B) CH₄ release rate from different sediment depths of thermokarst lakes BLH-A and BLH-B. The different colors represent the CH₄ release rates at the incubation temperatures of 5, 15, and 25°C. Values are means \pm standard errors (SE) (n=4).

Fig. S5. Temperature sensitivity (Q_{10}) of CH₄ release from different sediment depths of thermokarst lakes BLH-A and BLH-B. (A) BLH-A; (B) BLH-B.

Fig. S6. Relationships between the methane release rate and its temperature sensitivity (Q_{10}) and sediment age. (A) Relationship between the Q_{10} of CH_4 release and sediment age; (B) Relationship between the methane release rate and sediment age; (C) Changes in Q_{10} and methane release rates with sediment age.

Fig. S7. Relationships of cumulative CH₄ release with sediment properties, substrate availability, and microbial communities. SWC, sediment water content; SOC, soil organic carbon; TN, total nitrogen; POC, particulate organic carbon; MAOC, mineral-associated organic carbon; Bact, bacterial PLFAs; Act, actinomycetes PLFAs; F/B, the ratio of fungal PLFAs to bacterial PLFAs; S/V, the ratio of vanillyls to syringyls in ligninphenol; C/V, the ratio of vanillyls to cinnamyls in ligninphenol; The ratio of acid to aldehyde forms of vanillyls and syringyls [(Ad/Al)_v and (Ad/Al)_s]. The solid lines and grey area represent the linear regression lines and the 95% confidence interval, respectively. Circles and triangles denote BLH-A and BLH-B, respectively. R², the proportion of variance explained. *, ** and *** indicate significant correlation between CH₄ release and the corresponding variable at p<0.05, p<0.01, p<0.001, respectively.

3. The title of the manuscript and key point is that the Q₁₀ values of potential methane production from drained lakes is half of that of non-drained thermokarst lakes. Unfortunately, the manuscript lacks information about these other sites and the conditions under which Q₁₀ values were evaluated. With the current information, it is impossible for the readers to understand what these comparisons mean and it would be impossible to replicate results. Since this is the main takeaway of the paper, information on these sites and the study designs should be briefly touched on in the main text and the methods should include detailed information.

Response: Thank you for your good suggestion. We added a brief study design introduction in the main text and more detailed information for Q₁₀ values in the methods.

Lines 213-216: To reveal the changes of Q₁₀ affected by lake drainage, we conducted a preliminary comparison between the drainage-affected and non-drainage

thermokarst lakes by integrating the currently published and our unpublished Q_{10} data on the QTP based on the similar incubation experiments (Table S2; See methods).

Lines 498-507: Data synthesis of Q_{10} in thermokarst lakes

To show the effect of drainage on Q_{10} of CH_4 release in thermokarst lake sediments, we synthesized the published Q_{10} data of sediment CH_4 release in the non-drainage thermokarst lakes. Given large variations in CH_4 release from thermokarst lakes with different vegetation type²⁷, we just compiled the data from alpine meadow and wet meadow on the QTP. There is limited study reporting the Q_{10} of CH_4 release of thermokarst lake sediments²⁶. We compiled our unpublished Q_{10} data of non-drainage thermokarst lakes on the QTP based on similar laboratory incubation (Table S2). Considering the vertical variation of Q_{10} values in this study, the surface 0-30 cm data was used for the comparison.

4. Throughout the manuscript, especially in the results and discussion, there is a lack of comparisons to other drained lake basin studies in permafrost regions. The manuscript could also refer to other systems that dry and wet cyclically. Some examples are below:

a. Fuchs, M., Lenz, J., Jock, S., Nitze, I., Jones, B.M., Strauss, J., Günther, F. and Grosse, G., 2019. Organic carbon and nitrogen stocks along a thermokarst lake sequence in Arctic Alaska. *Journal of Geophysical Research: Biogeosciences*, 124(5), pp.1230-1247.

b. Skeeter, J., Christen, A., Laforce, A.A., Humphreys, E. and Henry, G., 2020. Vegetation influence and environmental controls on greenhouse gas fluxes from a drained thermokarst lake in the western Canadian Arctic. *Biogeosciences*, 17(17), pp.4421-4441.

c. Loiko, S., Klimova, N., Kuzmina, D. and Pokrovsky, O., 2020. Lake drainage in permafrost regions produces variable plant communities of high biomass and productivity. *Plants*, 9(7), p.867.

d. Walter Anthony, K., Daanen, R., Anthony, P., Schneider von Deimling, T., Ping, C.L., Chanton, J.P. and Grosse, G., 2016. Methane emissions proportional to permafrost carbon thawed in Arctic lakes since the 1950s. *Nature Geoscience*, 9(9), pp.679-682.

Response: We sincerely appreciate the valuable comments. We have checked the literature carefully and added them to show the effects of thermokarst lake on soil organic carbon, vegetation change and methane emissions into the Introduction parts in the revised manuscript.

For the effect of thermokarst lake on soil organic carbon stocks, Fuchs et al., (2019) has been cited in the introduction [**Lines:52-54** *CH_4 as a powerful greenhouse gas is produced in anaerobic environments, and drainage events can significantly change CH_4 release from thermokarst lakes by influencing sediment moisture, carbon decomposability¹⁸....*].

For the effect of thermokarst lake on vegetation changes, Skeeter et al., (2020) and Loiko et al., (2020) have been cited in the Introduction [**Lines:52-55** *CH_4 as a powerful greenhouse gas is produced in anaerobic environments, and drainage events can significantly change CH_4 release from thermokarst lakes by influencing sediment*

moisture, carbon decomposability¹⁸, and vegetation type markedly^{19,20,21}.].

For the effect of permafrost thaw on lake methane emissions, Walter Anthony et al., (2016) has been cited in the Introduction, which showing the importance of our study [**Lines:38-39** Thermokarst lakes serve as significant natural emission sources of methane (CH₄)^{5,6,7,8}.....].

5. The data availability statement is not enough. The original data generated by this study and also the literature review of Q₁₀ data should be provided in a csv format and stored in an open repository. It is not enough to say the data are available in the paper and supplemental, as the data are presented in figure form with no metadata. For the study to be reproducible, it is important to make data openly available.

Response: We have added TableS2 in Supplementary information to show literature review and unpublished of Q₁₀ data. Additionally, we have uploaded all data (including soil properties, carbon composition, microbial communities, functional gene abundance and community composition of methanogens, potential CH₄ release rates, and Q₁₀) to the figshare database. Here, we shared a private link (<https://figshare.com/s/2e26ea8ca49153d9ea0c>) for reviewers to review the Supporting Data. Additionally, we have also shared the relevant code for plotting and structural equation modelling via a private link (<https://figshare.com/s/7c7fba016f3266883454>). We have modified the data availability statement and added the code availability, as follows:

Lines 541-543: Data availability All data supporting the findings are available online in Supplementary information or the figshare data repository (<https://doi.org/10.6084/m9.figshare.26880526>)⁷³.

Lines 545-548: Code availability Data analysis was performed using R 4.1.3. which is publicly available at <https://www.r-project.org>. The supporting code is provided at Figshare (<https://doi.org/10.6084/m9.figshare.26892715>)⁷⁴.

Table S2 The published and our unpublished Q₁₀ data from the non-drainage thermokarst lakes on the QTP.

ID	Lake name	Vegetation type	Sediment depth (cm)	Incubation temp (°C)	Q ₁₀	Data source
1	BLH-1	Alpine wet meadow	0-20	5, 15	7.39	Unpublished
2	BLH-2	Alpine wet meadow	0-20	5, 15	1.27	Unpublished
3	BLH-4	Alpine meadow	0-20	5, 15	7.07	Unpublished
4	BLH-7	Alpine meadow	0-20	5, 15	12.77	Unpublished
5	BLH-11	Alpine wet meadow	0-20	5, 15	2.26	Unpublished
6	BLH-12	Alpine wet meadow	0-20	5, 15	6.68	Unpublished
7	BLH-13	Alpine meadow	0-20	5, 15	3.72	Unpublished
8	BLH-15	Alpine meadow	0-20	5, 15	3.83	Unpublished
9	BLH-18	Alpine wet meadow	0-20	5, 15	5.51	Unpublished
10		Alpine wet meadow	0-10	5, 20	21.85	Xu et al., 2024
11		Alpine meadow	0-10	5, 20	6.82	Xu et al., 2024

Minor comments:

Line 34: the term “aggravates” is not clear here as permafrost thaw can lead to thermokarst lake development and drainage.

Response: We have rewritten this sentence in lines 36-37 in the revised manuscript.

Lines 35-36: *Global warming causes the thawing of permafrost and accelerates the formation and drainage of thermokarst lakes.*

Line 38: the phrase “current warming and dry-induced permafrost thaw” is confusing here. I am not sure what it means. Please clarify.

Response: Thank you for pointing this out. It has been rewritten as follows:

Lines 42-44: *Notably, climate warming and permafrost thaw result in various drainage of thermokarst lakes and even drained events.*

Line 49: Is it not also possible that lake drainage could lead to wetland development and the presence of wetland vegetation that could lead to an increase in emissions compared to lake emissions?

Response: Good suggestion. We have modified and cited the related references as follows:

Lines 52-55: *CH₄ as a powerful greenhouse gas is produced in anaerobic environments, and drainage events can significantly change CH₄ release from thermokarst lakes by influencing sediment moisture, carbon decomposability¹⁸, and vegetation type markedly^{19,20,21}.*

Chen Y, Liu A, Cheng X. Vegetation grows more luxuriantly in Arctic permafrost drained lake basins. *Glob Chang Biol* 27, 5865-5876 (2021).

Loiko S, Klimova N, Kuzmina D, Pokrovsky O. Lake Drainage in Permafrost Regions Produces Variable Plant Communities of High Biomass and Productivity. *Plants* 9, (2020).

Skeeter J, Christen A, Laforce A-A, Humphreys E, Henry G. Vegetation influence and environmental controls on greenhouse gas fluxes from a drained thermokarst lake in the western Canadian Arctic. *Biogeosciences* 17, 4421-4441 (2020).

Lines 51- 55: Here the authors should be clear that the relationships they are discussing were demonstrated by prior studies. As it is currently written this is not quite clear.

Response: Thank you again for valuable suggestion. We have modified this sentence, as follows:

Lines 56-59: *“Additionally, it was shown that sediment carbon composition such as mineral-associated organic carbon (MAOC) contents increase in the drainage process due to the protection of flocculation, sorption, and co-precipitation²⁴.”*

Lines 101-103: To my knowledge, it is quite common for sediment carbon availability to be highest at the near surface of lake sediments and wetland sediments. Some discussion of this here is warranted to provide context to the findings.

Response: According to your great comments, we have revised the conclusion of this discussion in the revised manuscript.

Lines 113-115: *Taken together, our study demonstrates that higher substrate availability in surface sediment of thermokarst lakes is mainly dominated by POC compared to the deeper layers.*

Line 149: When talking about “CH₄ release” adding the word “potential” in front is warranted for this study since the CH₄ release values are from incubations and not field studies.

Response: We have carefully checked the manuscript and modified the descriptions.

Line 215: Throughout the paper “drives” should be “drivers”

Response: We are really sorry for our careless mistakes. Thank you for your reminder.

Line 314: What is a seasonal thermokarst lake? The characteristics of these lakes, including age and wetting dynamics, is a crucial component of the paper and should be discussed in the main body of the manuscript.

Response: We feel sorry for giving your confusion. In this study, the thermokarst lakes are referred to be dry and wet cyclically as seasonal drained thermokarst lakes. We have added the detailed description of the lakes shown in the above response [*lines 355, 364-379*].

Line 581: The data availability is not sufficient. Data files, including original study data and the literature Q₁₀ data, need to be provided csv form in an open repository.

Response: Thank you for your suggestion. We have provided the literature Q₁₀ data in Supplementary information and uploaded original study data to the figshare data repository in the above response.

Reviewer #2 (Remarks to the Author):

The authors obtained two thermokarst lake sediment cores in the drainage-affected alpine permafrost area of the central Qinghai-Tibet plateau, and combined the observations with a 150-day incubation experiment to reveal the temperature sensitivity and potential drivers of CH₄ production. The mitigating effect of drainage on CH₄ release from thermokarst lakes is well known (e.g., Helsop et al. 2020), yet, the effect of drainage on thermal sensitivity (Q₁₀ specifically) of methanogenesis and related processes has been less constrained. In this respect, the work provides interesting insights into parameterization of thermokarst processes of the central Qinghai-Tibet plateau. The consideration of microbial composition and biogeochemical processes underlying associated CH₄ release is equally interesting. The study is based on the sound methodology and in most cases provides sufficient detail for the work to be reproduced elsewhere. I believe that revision of the summarized below aspects of the manuscript would strengthen and highlight the most important conclusions.

Response: We sincerely appreciate your positive comments for our study. The

detailed comments are all valuable and very helpful for revising and improving our paper, as well as the important guiding significance to our future study. We have made careful revisions and hope our endeavor has fully addressed your comments.

-Introduction

Lines 31-42: More recent and more precise references could be used.

Response: Thank you for the valuable comment. We have checked the literature carefully and added more recent references into the Introduction part in the revised manuscript.

Mishra, U. et al. Spatial heterogeneity and environmental predictors of permafrost region soil organic carbon stocks. *Sci. Adv.* 7, eaaz5236 (2021).

Webb EE, Liljedahl AK. Diminishing lake area across the northern permafrost zone. *Nat Geosci* 16, 202-209 (2023).

Yang G, et al. Characteristics of methane emissions from alpine thermokarst lakes on the Tibetan Plateau. *Nat Commun* 14, 3121 (2023).

Li Y, et al. Methane Emissions From the Qinghai - Tibet Plateau Ponds and Lakes: Roles of Ice Thaw and Vegetation Zone. *Global Biogeochem Cycles* 38, (2024).

Brosius LS, Walter Anthony KM, Treat CC, Jones MC, Dyonisius M, Grosse G. Panarctic lakes exerted a small positive feedback on early Holocene warming due to deglacial release of methane. *Communications Earth & Environment* 4, (2023).

Lines 44-45: Please, indicate that discussed is the relevant Q_{10} meaning (e.g., 'here') rather than in general (in general the Q_{10} coefficient is a measure of the degree of temperature dependence of essentially any, e.g., chemical or biological, process).

Response: According to the comments, we have added the relevant meaning of Q_{10} in the revised version.

Lines 48-55: *Temperature sensitivity (Q_{10}) represents a key parameter of biogeochemical models that reflects the response of carbon release to warming^{13,14,15}. Quantifying the Q_{10} of CH_4 release is thus critical to improving CH_4 emissions assessments of thermokarst lakes and narrowing the uncertainty of permafrost carbon-climate feedback projections^{16,17}. CH_4 as a powerful greenhouse gas is produced in anaerobic environments, and drainage events can significantly change CH_4 release from thermokarst lakes by influencing sediment moisture, carbon decomposability¹⁸, and vegetation type markedly^{19,20,21}.*

Lines 52-55: There are many important contributors to CH_4 release; needs to clarify that these are rather a part of the total.

Response: Good suggestion. We have modified this sentence in the revised version.

Lines 59-60: *These influencing factors are the main determinants of CH_4 release from thermokarst lakes^{25,26,27}.*

Lines 55-57: Water content/soil moisture is at least equally important a factor; especially over a much shorter temporal scale compared to that of a 10C T change

potential; sediment exposure (i.e., O₂ availability) is equally important.

Response: We have added the relevant description to show the importance of changes in sediment moisture in the above response [**Lines 53-55**].

Lines 60: The ref 17 seems to have a slightly different focus (also discussed in lines 61-63), while itself rather pointing to the cited ("Third Pole" (TP), air temperature increase studies by Cheng et al., 2019, Cao et al., 2018; Wu et al., 2015; Wu et al., 2010, and Kuang and Jiao, 2016; Pang et al., 2012; Yang et al., 2019); perhaps another reference would be more suitable here.

Response: According to your careful comments, we have updated the ref 28 as follows:

Yan, Y., You, Q., Wu, F. et al. Surface mean temperature from the observational stations and multiple reanalyses over the Tibetan Plateau. Clim Dyn 55, 2405–2419 (2020). <https://doi.org/10.1007/s00382-020-05386-0>.

-Results

Line 121: Please, highlight the interpretation and significance of the observed variation.

Response: Thank you for the valuable comment. We have rewritten this sentence.

Lines 132-135: *The abundances of all microbial groups, including bacterial, fungal, and actinomycetes PLFAs, significantly decrease with depth (Fig. S1 and 2). The higher microbial abundance in surface 0-30 cm depth (all $p < 0.01$; Fig. 2B) could be attributed to higher moisture and substrate availability in surface sediment.*

Lines 128-129: Please, rephrase for clarity.

Response: We feel sorry for your confusion, we have rewritten this sentence.

Lines 146-149: *Regarding the methanogenic community composition of sediment from drainage-affected thermokarst lakes, there were three dominant methanogenic orders, including Methanomicrobiales, Methanobacteriales, and Methanosarcinales (Fig. S3).*

Line 130: Also, there seems to be a very interesting set of taxonomic and biochemical data available, yet very little discussion in the text of their significance (mostly in Figures- Fig 2, 4, 5, S3, S7), nor the discussion of the associated methanogenesis pathways.

Response: Thank you for your great suggestion. We have added the related discussion in the revised Results and discussion.

Lines 146-161: *Regarding the methanogenic community composition of sediment from drainage-affected thermokarst lakes, there were three dominant methanogenic orders, including Methanomicrobiales, Methanobacteriales, and Methanosarcinales (Fig. S3). This finding is consistent with results from non-drainage thermokarst lake sediments on the QTP⁵ and drainage-affected permafrost tundra in the Arctic³⁴. The order composition diagrams showed that Methanomicrobiales and Methanobacteriales were the most abundant methanogenic orders, representing 88-99% of all methanogens. It has been reported that the orders Methanomicrobiales and*

Methanobacteriales take carbon dioxide (CO₂) plus hydrogen (H₂) as substrates to produce CH₄³⁵. The results reveal that the CH₄ production pathway of drainage-affected thermokarst lakes can predominantly be hydrogenotrophic, which is consistent with a recent study about non-drainage thermokarst lakes on the QTP based on stable carbon isotope ($\delta^{13}\text{C}$) of CH₄ and CO₂⁵. This is possibly attributed to sediment organic matter affected by waterlogging before lake drainage is not completely degraded, providing the substrates for CH₄ production such as benzoate, CO₂, and H₂³⁶.

Line 131: Better perhaps to say - ... 'confirms' previous study.

Response: According to the comments, we have added the related discussion shown in the above response [**Lines 149-151**].

Lines 132-134: = different methanogenesis pathways.

Response: Thank you for your kind comments. We have added the related description about methane production pathways shown in the above response [**Lines 151-161**].

-Q10

Lines 149-150: The two cores appear to have very different soil compositions at similar depths; this may be a confounding variable in this case, perhaps worthy of some consideration and/or discussion. Also, Fig S4 - perhaps, present on a log scale? (Also Fig S6 - B).

Response: According to your valuable comments, we have added the discussion in the revised manuscript. Additionally, the Fig S6 has been redrawn in the revised version. We tried to use a log scale to show Fig S4, but it didn't seem well shown. Thus, we retain the original presentation of Fig S4.

Lines 186-189: *By contrast, the cumulative CH₄ release was higher in BLH-B than in BLH-A, especially in the surface 0-100 cm, which may be due to large differences in moisture content, SOC and TN content, and microbial abundance in the sediment cores.*

Line 155: Why the 30 cm layer exactly? Do you by chance have the moisture/O₂ content profiles for the sites?

Response: We measured the sediment moisture contents of two thermokarst lakes, which are shown in Fig. 1 C and D. In the surface 30 cm, sediment moisture content is highest compared to other depths. Below the surface 30 cm, Q₁₀ decreases rapidly, especially in BLH-B core.

Lines 168-177: Would be interesting to see some discussion of potentially confounding variables here (e.g., the carbon age depth profile, moisture, O₂ availability, etc.)

Response: We have added discussion to present the potentially confounding variables.

Lines 93-96: *Sediment moisture content rapidly decreases along the depth profile and has the highest value of 14-69% in the surface 0-30 cm (Fig. 1C and D). Compared to saturated conditions in non-drainage thermokarst lakes, lake drainage*

significantly reduces sediment moisture contents and increases oxygen (O₂) availability.

Lines 99-101: *The carbon ages of lake sediments increase gradually with depth, which indicates that surface sediment has a younger carbon age and shorter time of carbon turnover.*

Additionally, we also discussed the effect of O₂ on temperature sensitivity of CH₄ release in the section of “Temperature sensitivity of CH₄ release” [**Lines 224-233**].

Lines 179-183: check methods?

Response: We have added more details for Q₁₀ values in the Methods shown in the above response [**Lines 498-507**].

Lines 183-185: a similar result would be expected due to lower moisture content (if this was the case in drainage affected lakes); please, clarify, and highlight the importance of your parameterization of the process, which is indeed interesting.

Response: Thank you for the acknowledgment and comments. We have added a sentence to highlight the importance of parameterization of the process.

Lines 219-221: *This result ties well with previous studies, showing lower water table can cause a decrease in the temperature dependence of CH₄ emissions in wetland ecosystems⁴⁰.*

Lines 222-224: *This result suggests that low temperature sensitivity of drainage-affected thermokarst lakes is essential for assessing CH₄ emissions from thermokarst lakes.*

Lines 186-187: There may be other (on top of the methanogens survival) factors influencing CH₄ release in these conditions, e.g., biochemical.

Response: Sorry for the confusion. We have rephrased this sentence and explained the decrease of Q₁₀ in the drainage-affected thermokarst lakes from two aspects.

Lines 222-234: *This result suggests that low temperature sensitivity of drainage-affected thermokarst lakes is essential for assessing CH₄ emissions from thermokarst lakes. There are two possible reasons for the low temperature sensitivity. First, thermokarst lake drainage dramatically alters oxygen conditions and hydrological³, threatening the survival of methanogens. On the one hand, thermokarst lake drainage exposes surface sediments to the atmosphere, resulting in the death of methanogens due to oxygen toxicity⁴¹. On the other hand, lake drainage reduces the water table depth and increases water stress on microorganisms, thereby decreasing methanogenic activity⁴¹. Hence, the changes in O₂ availability and moisture contents are important in revealing the processes of CH₄ production and oxidation in drainage-affected thermokarst lakes and may be a key indicator for future modelling. Second, lake drainage can lead to the binding of labile carbon to minerals, reducing substrate availability and thus inhibiting CH₄ release⁴².*

Lines 193-194: Would be interesting to see some more details and clarification here.

Response: According to your great comments, we have added more details as follows.

Lines 237-240: *Abrupt permafrost thaw reduces soil moisture and improves aeration, allowing oxygen to replace Fe(III) as electron acceptors for microbial respiration and stimulating Fe(II) oxidation to Fe(III), thereby promoting the formation of iron-bound organic carbon^{44,45}.*

Line 195: This is a very interesting conclusion (halved Q_{10}); was it based on incubation rather than observation? were there any in-situ measurements?

Response: Good comments. This is the result of incubation experiment due to limited observation of methane emissions from the drainage-affected thermokarst lakes. Thus, future related field in-situ measurements are urgently needed. We hope our study sheds light on the crucial effects of lake drainage on the terrestrial carbon cycle.

Lines 213, 432: Please, capitalize and/or italicize SE.

Response: Thanks for your careful checks. We have capitalized SE in the revised manuscript.

Lines 219-227: Perhaps highlighting the quantitative aspects here would be more interesting than the facts themselves.

Response: According to the valuable comments, we have rewritten this paragraph as follows.

Lines 269-278: *The results show that the cumulative CH₄ release is closely related to the sediment properties and substrate availability. Specifically, cumulative CH₄ release positively correlates with SWC, SOC, and TN (all $p < 0.001$), and negatively correlates with clay content, (Ad/Al)_v, and (Ad/Al)_s (all $p < 0.05$; Fig. S7). Higher microbial abundance (all $p < 0.001$) and mcrA gene abundance ($R^2 = 0.71$, $p < 0.001$) are associated with higher CH₄ release (Fig. S7). Likewise, substrate availability and microbial communities are also the key predictors of Q_{10} variations (Figs. 4 and 5). Specifically, the Q_{10} of CH₄ release is positively correlated with microbial abundance, SOC, TN, and SWC (all $p < 0.01$); while is negatively correlated with (Ad/Al)_v ($R^2 = 0.27$, $p < 0.05$; Fig. 4).*

Line 230: TN may be either or both, contributor and/or an indicator; please clarify; please, also, rephrase 'badly' influenced.

Response: Thank you for your careful comments. TN is an important contributor to CH₄ release. What's more, we have replaced "driver" with "contributor" in the revised manuscript. Meanwhile, we have replaced "badly" with 'severely'.

Lines 280-283: *Thus, our result suggests that TN is a crucial contributor to CH₄ release from thermokarst lakes. For lake drainage events, sediment moisture is severely influenced due to the changes from a waterlogged environment to terrestrial ecosystems.*

Lines 233-234: Please, rephrase for clarity.

Response: Sorry for the confusion. We have modified this sentence in the revised

manuscript.

Lines 285-286: *Consistent with this deduction, we find faster CH₄ release in surface sediment with higher moisture contents (Fig. 1 and 3).*

Lines 235-236: Please, clarify, whether the clay content or simply less carbon [content] was considered here.

Response: Thank you for your careful comments. It has been revised as follows:

Lines 288-290: *Soil organic carbon can be stabilized by chemical interaction with clay minerals by forming organic-mineral and physical occlusion within microaggregates⁴⁶.*

Lines 256-260: CH₄ release

Response: Thank you for your suggestion. We have modified this sentence:

Lines 314-315: *This result is consistent with the CH₄ release in non-drainage thermokarst lakes on the QTP²⁷.*

Lines 260-264: The presented SEM attributes 74.3\% variation in Q₁₀ to carbon availability and microbial properties; hence, what about N₂ (that seems to be included in C availability as per Fig 5), and also water and O₂ availability? Also, am I understanding correctly that these were calculated based on wet incubations in anaerobic conditions? Could you, perhaps, add some details (in the text or Supplement) to clarify your findings.

Response: Thank you for your valuable comments. In this study, we conducted an anaerobic laboratory incubation using fresh sediments through field sampling. We did not consider field environment factors such as N₂ and O₂ availability. To avoid any confusion, we have added the information in Methods. Additionally, we have added the discussion in the Results and discussion.

Lines 230-233: *Hence, the changes in O₂ availability and moisture contents are important in revealing the processes of CH₄ production and oxidation in drainage-affected thermokarst lakes and may be key indicators for future CH₄ modelling.*

Lines 281-282: Please, rephrase for clarity.

Response: Sorry for the confusion. We have rephrased this sentence in the revised manuscript.

Lines 335-337: *Variation partitioning modelling evaluates the relative contributions of sediment properties, substrate availability, and microbial communities in explaining variations in CH₄ release and Q₁₀ variation.*

Lines 290-294: very important point indeed, especially providing many possible interlinkages between CH₄ and CO₂ emissions.

Response: Thanks. We will further explore the effects of drainage on CH₄ and CO₂ emissions for thermokarst lakes combined with field monitoring and laboratory incubation.

-Methods

Line 301: Does active layer - 1.5 to 2.5 m - affects the cores? (BLH-A - 3.87 m, BLH-B- 5.55 m; not discussed in Results).

Response: Thank you for the suggestion. The depth range of 1.5 to 2.5 m represents the thickness of active layer across the Beilu River Basin. The two thermokarst lakes are close to each other, and the active layer thickness had little effect on sediment cores.

Lines 310-312: Please, rephrase for clarity.

Response: Sorry for the confusion. We have rewritten this sentence and added more detail about thermokarst lakes in study regions shown in the above response (**Lines 364-379**).

Line 315: Please add a brief description of the drilling rig, e.g., in the Supplement.

Response: Thank you for your great comment. We have added the description about the drilling rig in the Supplement and provided a photo from the drilling.

Lines 25-30:

Supplementary Text 1: Drilling methods

We used an exploration drill rig (Wendeng GJ-240, China) with a single core barrel to sample the sediment cores of thermokarst lake. The rig is self-propelled by tracks and has a weight of around 750 kg. The hydraulic rig, powered by a diesel engine, and uses rotary core drilling methods. The rig has a capability of 420 m with a wireline system.

Fig. S9 Sediment core sampling of thermokarst lakes using a drilling rig.

Line 401: Please identify the 'previous study' here; also, did your team participate in that study?

Response: To avoid any confusion, we have modified this sentence as follows:

Lines 469-470: *The potential CH₄ release rates were measured using the method described by Heslop et al.³⁶.*

Line 412: Please, specify whether CH₄ concentrations were measured directly in the headspace or sampled/extracted and then measured?

Response: Thanks for the detailed comments. We sampled 1 mL headspace gas with a syringe and injected it into the Gas chromatograph to measure CH₄ concentration. To avoid confusion, we have added more details in the revised Methods.

Lines 479-482: *To measure CH₄ concentration, we sampled 1 mL headspace gas with a syringe and injected it into the gas chromatograph (GC, Agilent-7890A). We measured the changes in CH₄ production at nine time points during a 150-day incubation.*

Lines 431, 433, 437, 439, 442, 446, 459, 460: Please acknowledge/cite R and R packages (info may be found e.g., in R itself as `citation()` and `citation("package_name")` respectively)

Response: Following your comments, we have cited the relevant reference in the corresponding position.

Additional reference:

Team RC. A Language and Environment for Statistical Computing. R Foundation for Statistical Computing, Vienna, Austria (2024).

Wickham H. ggplot2: Elegant Graphics for Data Analysis. Springer-Verlag New York (2016).

Fox. J WS. An R Companion to Applied Regression, Third edition. Sage, Thousand Oaks CA (2019).

Kassambara A. Pipe-Friendly Framework for Basic Statistical Tests. R package version 0.7.2 (2023).

-Supplement

Fig S2: What is the different colours meaning?

Response: Sorry for your confusion. We have added the notes in revised Supplementary Information.

Lines 50-51: Box represents the interquartile range, with red and blue indicating surface 0-30 cm layer and deeper >30 cm layer, respectively.

Fig S4: Perhaps a log scale could be considered.

Response: Thank you for your suggestions. We tried to use a log scale, but it didn't seem well shown. Thus, we retain the original presentation of Fig S4.

Fig S6 - B: Perhaps a log scale could be considered.

Response: Good suggestion. We have modified Fig S6-B using a log scale in the revised version.

REVIEWERS' COMMENTS

Reviewer #1 (Remarks to the Author):

I appreciate the authors' thorough responses and updates to the initial reviewer suggestions. I appreciate the in-depth analysis performed by the authors, including the use of biogeochemical and microbial approaches to identify the controls on the temperature sensitivity of methane production in the sediments from these lakes. Their findings that methane production from seasonally drained thermokarst lakes has a lower temperature sensitivity than non-drainage affected thermokarst lakes is novel and would be an important contribution towards improving permafrost carbon emission models (however, see my second point below about the non-drainage affected lakes).

The authors have adequately updated their methods and now include relevant data tables and links to the data. They have also made a great effort to clarify their reasoning in many places. I have a few minor comments I hope the authors can address.

First, I still think the description of the study lakes can be improved. The thermokarst lakes are referred to as "drainage lakes", but the lakes are not permanently drained- they drain seasonally. On line 43, I suggest the authors mention that one impact of climate warming and permafrost thaw is the development of seasonally drained lakes. I also think it would be helpful to know how abundant seasonally drained lakes are compared to lakes that drain more permanently.

Second, the comparison between the q10 values from the "drainage-affected" (study lakes) and the unpublished "non-drainage thermokarst lakes" is still lacking clarity. In the methods, the authors refer to the "non-drainage lakes" as alpine meadows and wet meadows. This is confusing because this definition suggests they are permafrost wetlands and meadows, and not thermokarst lakes. If these systems are thermokarst lakes then I suggest the descriptions are updated for clarification. If these systems are not lakes, then the description and naming should be changed to highlight they are wetlands.

Finally, the shared code could be updated with more detail, including a readMe file and examples of the code with the datafiles also shared to help with reproducibility.

Reviewer #1 (Remarks on code availability):

The code does not include a readMe file. The code is for figures and the SEM, but it appears that many of the other types of analyses in the paper are left out (for example, many of the statistical analyses). The code is also not set up to run with the data files and tables provided. Instead, the code just names the data as "rawdata". This makes it difficult to reproduce the analyses. I would suggest providing more detailed code that includes the file names of the files provided so readers can more easily re-run analyses.

Reviewer #2 (Remarks to the Author):

The authors made a thorough revision of the manuscript and addressed all raised questions; thank you. This revision substantially clarifies the study design, methodology, and key findings. Few final points could, I believe, benefit from some more discussion now:

- (1) The BLH-A site appears lacking the young carbon at the surface - is there any explanation for this?
- (2) Thank you for adding more details on the methodology of incubations; perhaps adding in the supplement some data on measured CH₄ concentrations would be useful as well (i.e., undiluted, the 1 mL sample size seems quite small a volume for the reliable GC analysis); please, also note in the methods if the extracted 1 mL volume was somehow compensated in the incubation bottles (else, discuss the potentially reduced headspace volume over time);
- (3) Please, also clarify if for each sample the T was raised during the incubation, or the replicate samples were incubated at different T for 150 days simultaneously.

Minor details:

line 42-44: please, rephrase for clarity if necessary; else, the notion seems already indicated in lines 35-36;

lines 48-52 and 52-55 : as a suggestion, I would reverse the presentation order of the two notions for segueing.

Reviewer #2 (Remarks on code availability):

I did not run the code - missing data.

Response to Reviewers

Reviewer #1 (Remarks to the Author):

I appreciate the authors' thorough responses and updates to the initial reviewer suggestions. I appreciate the in-depth analysis performed by the authors, including the use of biogeochemical and microbial approaches to identify the controls on the temperature sensitivity of methane production in the sediments from these lakes. Their findings that methane production from seasonally drained thermokarst lakes has a lower temperature sensitivity than non-drainage affected thermokarst lakes is novel and would be an important contribution towards improving permafrost carbon emission models (however, see my second point below about the non-drainage affected lakes). The authors have adequately updated their methods and now include relevant data tables and links to the data. They have also made a great effort to clarify their reasoning in many places. I have a few minor comments I hope the authors can address.

Response: We sincerely appreciate the valuable comments. We further revise the manuscript carefully, and the detailed responses are shown as follows:

First, I still think the description of the study lakes can be improved. The thermokarst lakes are referred to as "drainage lakes", but the lakes are not permanently drained- they drain seasonally. On line 43, I suggest the authors mention that one impact of climate warming and permafrost thaw is the development of seasonally drained lakes. I also think it would be helpful to know how abundant seasonally drained lakes are compared to lakes that drain more permanently.

Response: We have added the related information for seasonally thermokarst lakes in the revised manuscript.

Lines 36-38: *"Global warming causes the thawing of permafrost and accelerates the formation, expansion, and drainage (seasonal, intermittent, or permanent) of thermokarst lakes^{3,4,5}."*

Lines 40-44: *"Thermokarst lakes formation and expansion increase methane (CH₄) emission as accelerated permafrost thaw beneath and around lakes unlocks previously frozen sediments for microbial anaerobic decomposition^{6,7}. Various studies have shown that thermokarst lakes serve as significant natural emission sources of CH₄^{6,8,9,10}, ..."*

Second, the comparison between the q10 values from the "drainage-affected" (study lakes) and the unpublished "non-drainage thermokarst lakes" is still lacking clarity. In the methods, the authors refer to the "non-drainage lakes" as alpine meadows and wet meadows. This is confusing because this definition suggests they are permafrost wetlands and meadows, and not thermokarst lakes. If these systems are thermokarst lakes then I suggest the descriptions

are updated for clarification. If these systems are not lakes, then the description and naming should be changed to highlight they are wetlands.

Response: We have corrected it. The revised sentence is shown as follows:

Lines 512-514: *“Given large variations in CH₄ release from thermokarst lakes with different vegetation type²⁷, we only compiled the Q₁₀ data on CH₄ release from lake sediments located in alpine meadow and wet meadow regions on the QTP.”*

Finally, the shared code could be updated with more detail, including a readMe file and examples of the code with the datafiles also shared to help with reproducibility.

Response: Yes. The updated code including a readme file and examples of the code with the datafiles. We have added the source data and code.

Reviewer #1 (Remarks on code availability):

The code does not include a readMe file. The code is for figures and the SEM, but it appears that many of the other types of analyses in the paper are left out (for example, many of the statistical analyses). The code is also not set up to run with the data files and tables provided. Instead, the code just names the data as "rawdata". This makes it difficult to reproduce the analyses. I would suggest providing more detailed code that includes the file names of the files provided so readers can more easily re-run analyses.

Response: We have updated the source data, code and readme file as shown in the above response.

The private links (<https://figshare.com/s/2e26ea8ca49153d9ea0c>) (<https://figshare.com/s/7c7fba016f3266883454>) are provide for reviewers to review the Supporting Data and relevant code.

Reviewer #2 (Remarks to the Author):

The authors made a thorough revision of the manuscript and addressed all raised questions; thank you. This revision substantially clarifies the study design, methodology, and key findings. Few final points could, I believe, benefit from some more discussion now:

(1) The BLH-A site appears lacking the young carbon at the surface - is there any explanation for this?

Response: Thank you for your valuable review. There are two possible reasons for older carbon of surface sediments at BLH-A. First, the surface area of BLH-A is about five times that of BLH-B, which results in a slower deposition rate. Second, the BLH-A lake is located in the alpine meadow with lower vegetation productivity around and within lake compared to the BLH-B lake, which led to a slower depositional process.

(2) Thank you for adding more details on the methodology of incubations; perhaps adding in the supplement some data on measured CH₄ concentrations would be useful as well (i.e., undiluted, the 1 mL sample size seems quite small a volume for the reliable GC analysis); please, also note in the methods if the extracted 1 mL volume was somehow compensated in the incubation bottles (else, discuss the potentially reduced headspace volume over time);

Response: We provided the data of measured CH₄ concentrations and uploaded it to the Figshare database. Additionally, we have added the related information in the Methods as follows:

Lines 491-492: *“Subsequently, 1 mL of N₂ was added to the bottles to maintain air pressure equilibrium in the bottle.”*

(3) Please, also clarify if for each sample the T was raised during the incubation, or the replicate samples were incubated at different T for 150 days simultaneously.

Response: We have clarified the methods of incubation experiment in the revised manuscript as follows:

Lines 485-489: *“For each sample, the replicate samples were incubated anaerobically at 5 °C, 15 °C, and 25 °C in the dark simultaneously. The incubation temperature represents the mean annual temperature, maximum summer temperature, and warming conditions at the bottom of the thermokarst lake on the QTP, respectively⁶⁵.”*

Minor details:

line 42-44: please, rephrase for clarity if necessary; else, the notion seems already indicated in lines 35-36;

Response: We have rewritten it as follows:

Lines 36-38: *“Global warming causes the thawing of permafrost and accelerates the formation, expansion, and drainage (seasonal, intermittent, or permanent) of thermokarst lakes^{3,4,5}.”*

Lines 47-49: *“Conversely, thermokarst lake drainage intensely alters hydrological dynamics⁴, potentially affecting CH₄ release to the atmosphere³.”*

lines 48-52 and 52-55: as a suggestion, I would reverse the presentation order of the two notions for segueing.

Response: Yes, we have reversed it in the revised version as follows:

Lines 53-65: *“CH₄ as a powerful greenhouse gas is produced in anaerobic environments, and drainage events can significantly change CH₄ release from thermokarst lakes by influencing sediment moisture, carbon decomposability¹³, and vegetation type markedly^{14,15,16}. Influenced by drainage events, microbial abundance and methanogenic communities of lake sediments had great changes^{18,19}. Additionally, it was shown that sediment carbon composition such as mineral-associated organic carbon (MAOC) contents increases in the*

drainage process due to the protection of flocculation, sorption, and co-precipitation²⁰. These influencing factors are the main determinants of CH₄ release from thermokarst lakes^{21,22,23}. Temperature sensitivity (Q₁₀) represents a key parameter of biogeochemical models that reflects the response of carbon release to warming^{17,18,19}. Quantifying the Q₁₀ of CH₄ release is thus critical to improving CH₄ emissions assessments of thermokarst lakes and narrowing the uncertainty of permafrost carbon-climate feedback projections^{20,21}."

Reviewer #2 (Remarks on code availability):

I did not run the code - missing data.

Response: We further added the source data and code. The raw data underlying each figure available as source data files is also provided.

The updated data and code were uploaded the Figshare database. Here, we shared a private link (<https://figshare.com/s/2e26ea8ca49153d9ea0c>) for reviewers to review the Supporting Data. Additionally, we have also shared the relevant code for plotting, structural equation modelling, multiple linear regression, and hierarchical partitioning via a private link (<https://figshare.com/s/7c7fba016f3266883454>).